# IN-CONTEXT COMPOSITIONAL Q-LEARNING FOR OFFLINE REINFORCEMENT LEARNING

**Qiushui Xu**[1], **Yuhao Huang**[2], **Yushu Jiang**[3], **Lei Song**[4], **Jinyu Wang**[4], **Wenliang Zheng**[1], **Jiang Bian**[4]

[1] Penn State University, [2] Nanjing University, [3] University of Toronto, [4] Microsoft Research
```
{qjx5019,wmz5132}@psu.edu, huangyh@smail.nju.edu.cn,
barryy.jiang@mail.utoronto.ca,
{Lei.Song,Wang.Jinyu,Jiang.Bian}@microsoft.com
```

## ABSTRACT

Accurate estimation of the Q-function is a central challenge in offline reinforcement learning. However, existing approaches often rely on a shared global Q-function, which is inadequate for capturing the compositional structure of tasks that consist of diverse subtasks. We propose In-context Compositional Q-Learning (`ICQL`), an offline RL framework that formulates Q-learning as a contextual inference problem and uses linear Transformers to adaptively infer local Q-functions from retrieved transitions without explicit subtask labels. Theoretically, we show that, under two assumptions—linear approximability of the local Q-function and accurate inference of weights from retrieved context—`ICQL` achieves a bounded approximation error for the Q-function and enables near-optimal policy extraction. Empirically, `ICQL` substantially improves performance in offline settings, achieving gains of up to 16.4% on kitchen tasks and up to 8.8% and 6.3% on MuJoCo and Adroit tasks, respectively. These results highlight the underexplored potential of in-context learning for robust and compositional value estimation and establish `ICQL` as a principled and effective framework for offline RL.

## 1 INTRODUCTION

Offline reinforcement learning (Offline RL) aims to learn effective policies from fixed datasets without further interaction with the environment (Fujimoto et al., 2019; Lange et al., 2012). This setting is especially important in real-world domains such as robotics (Kalashnikov et al., 2018), logistics (Wang et al., 2021), and operations research (Hubbs et al., 2020; Mazyavkina et al., 2021), where environment access is limited, data collection is expensive or risky, and historical data is often the only available resource. A central challenge is distributional shift: when a learned policy queries state-action pairs outside the dataset support, value extrapolation can cause severe overestimation and degenerate performance. (Fu et al., 2020; Kumar et al., 2020)

Contemporary methods primarily employ policy constraints (Chen et al., 2021) or value regularization (Kumar et al., 2020; Kostrikov et al., 2022) to address this challenge. However, policy-constraint methods are largely limited by the behavior policies used to collect the offline data and exhibit a trade-off between generalization and safe adherence to the constraint. Recent value-regularization methods aim to provide conservative estimates that impose softer penalties on out-of-distribution actions. Nevertheless, the optimality of the learned value function is not guaranteed when the static dataset is limited and potentially biased.

We observe that, in many RL control tasks, the state space can often be naturally divided into multiple subtasks. Although an expressive action-value function may in principle capture state-action values accurately, the knowledge learned from one subtask may not be fully transferable to another. For example, in MuJoCo locomotion tasks, knowledge for increasing walking speed may not help with recovery from unexpected non-nominal states. A visualization of this phenomenon is provided in Figure 1, which shows the distribution of states after dimensionality reduction, with colors indicating their actual future returns in the offline dataset. Moreover, although states in the dataset can often be grouped into coherent clusters, each of which typically corresponds to a specific subtask,

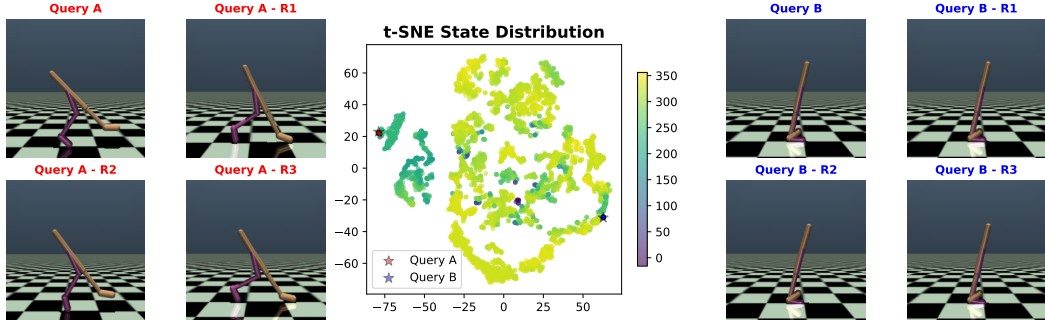

Figure 1: Center: dimension-reduced states and SAC value estimates on `Walker2d-Medium-Expert`. Left and right: two groups of similar states.

two clusters that are geometrically similar may nevertheless correspond to semantically different behaviors and exhibit distinct long-horizon returns. When offline data are insufficient and exploration is unavailable, this property is not naturally captured by an offline value-learning algorithm that fits a shared global value function.

To address these challenges, we propose to cast value learning in offline reinforcement learning as a contextual inference problem, thereby enabling local Q-function approximation through in-context learning. Specifically, we introduce In-context Compositional Q-Learning (`ICQL`), a general framework for offline RL that leverages the in-context learning capability of linear Transformers to infer local Q-functions from small retrieved sets of transitions. Rather than fitting a global approximator of the value function, `ICQL` leverages the compositional nature and local structure of the task to learn a family of value functions, thereby enabling flexible local adaptation of value estimation within context windows. Our key contributions are summarized as follows:

- To the best of our knowledge, we introduce the first offline RL framework `ICQL` that **formulates Q-learning as a contextual inference problem**, leveraging in-context learning with linear Transformers to adaptively infer local Q-functions without requiring explicit subtask labels or predefined subtask structure.

- We provide a theoretical analysis showing that **`ICQL` achieves bounded approximation error** under two assumptions—linear approximability of the local Q-function and accurate inference of the weights from retrieved context—and prove that the greedy policy with respect to the inferred Q-function is **near-optimal**.

- **`ICQL` improves performance in offline settings through in-context local approximation**, and we demonstrate the effectiveness of `ICQL` in both offline Q-learning and offline actor-critic frameworks. On Gym and Adroit tasks, `ICQL` yields score improvements of **8.8%** and **6.3%**, respectively. Notably, on Kitchen tasks, `ICQL` achieves a **16.4%** performance improvement over the second-best baseline. We further show that `ICQL` yields more accurate value estimation than strong baselines. These results highlight the underexplored potential of linear attention for robust and compositional value estimation in offline RL.

- We conduct extensive ablation studies to isolate the contributions of in-context learning and localized value inference. In addition, we investigate the impact of different retrieval strategies, including similarity metrics and context selection criteria, on overall performance and stability.

## 2  RELATED WORK

**Offline Reinforcement Learning.** Offline RL aims to learn effective policies from static datasets without further interaction with the environment. A central challenge in this setting is distributional shift, which can lead to severe value overestimation and degraded policy performance when the learned policy queries out-of-distribution actions. Several influential approaches address this issue by modifying Q-learning objectives or introducing conservative regularization. Representative examples include `CQL` (Kumar et al., 2020), `IQL` (Kostrikov et al., 2022), and `TD3+BC` (Fujimoto & Gu, 2021). `CQL` introduces a conservative penalty on Q-values for out-of-distribution actions to mitigate

overestimation. `TD3+BC` combines TD3 with a behavior cloning loss to bias policy updates toward actions in the dataset while retaining value-based learning. `IQL` removes explicit policy optimization and learns value-weighted regression targets to implicitly extract high-value actions from offline data. More recent work has continued to improve offline RL from several complementary directions. `ReBRAC` (Tarasov et al., 2023) revisits the minimalist actor-critic design in offline RL and shows that strong performance can be obtained through careful regularization and implementation choices. `DMG` (Mao et al., 2024) studies how to obtain better generalization in offline RL under mild assumptions. `FQL` (Park et al., 2025) introduces a flow-based perspective for Q-learning, while `QC` (Li et al., 2025) explores temporally structured action representations through action chunking. In addition, retrieval-augmented sequence-modeling approaches such as `RA-DT` (Schmied et al., 2024) enrich decision-transformer-style policies with external memory for in-context RL. Despite their differences, these methods still mainly rely on global value or policy models trained over the entire dataset. Such global modeling can be limiting in compositional environments, where local transition structure and subtask-specific value patterns may vary substantially across the state space. In contrast, our approach casts value learning as a collection of local estimation problems and uses in-context inference to adapt Q-functions to retrieved local transition dynamics without requiring additional supervision.

**In-context Learning in RL.** Recent work has applied Transformers to offline RL, often through sequence modeling for return-conditioned policy learning (Zhao et al., 2025). For example, Decision Transformer (Chen et al., 2021) and Gato (Reed et al., 2022) treat trajectories as sequences, while replay-based in-context RL (Chen et al., 2021; Reed et al., 2022) uses Transformers for behavior cloning and reward learning. These approaches leverage the ability of pre-trained Transformers to adapt through prompt conditioning or in-context learning. In-context learning has both a strong theoretical foundation (Von Oswald et al., 2023; Shen et al., 2024; Wang et al., 2025b) and strong empirical performance across tasks (Hollmann et al., 2023; Micheli et al., 2023), and it is increasingly studied in supervised settings (Laskin et al., 2023; Lee et al., 2023; Mukherjee et al., 2025). (Laskin et al., 2023) proposes Algorithm Distillation (`AD`) to mimic the data-collection policy, but this approach is constrained by the quality of the original algorithm. `DPT` (Lee et al., 2023) improves regret in contextual bandits through in-context learning, but it assumes access to optimal actions, which is often unrealistic in offline RL. `PreDeToR` (Mukherjee et al., 2025) adds reward prediction to Decision Transformer models, yet it still focuses on action generation. While these approaches focus on directly generating actions or policies from trajectories, they do not explicitly target value estimation, which is outside the scope of this paper. Therefore, we do not include these methods as baselines. Although recent work has explored Transformers in offline RL primarily for trajectory modeling or return-conditioned generation (Chen et al., 2021; Laskin et al., 2023; Mukherjee et al., 2025), we instead study linear attention as a tool for in-context value learning. Our results suggest that linear attention, when used for local Q-function estimation, provides strong performance and generalization benefits. To our knowledge, this is the first work to demonstrate the potential of linear attention for compositional value-based offline RL.

## 3 METHODOLOGY

In this section, we introduce the proposed `ICQL` framework, including its local value modeling, retrieval mechanism, learning procedure, and theoretical properties.

### 3.1 LOCAL Q-FUNCTIONS

In this section, we define local Q-functions for offline RL based on the local neighborhood associated with each state. We let $\mathcal{D}$ denote the dataset containing all offline transitions.

**Definition 3.1.** (Local $Q$-function Approximation) Given a transition $(s, a, r, s', a') \in \mathcal{D}$, for some $d, \bar{d} > 0$, a nearby transition $(\bar{s}, \bar{a}, \bar{r}, \bar{s}', \bar{a}') \in \mathcal{D}$ is defined as

$$(\bar{s}, \bar{a}, \bar{r}, \bar{s}', \bar{a}') \in \left\{ (s_i, a_i, r_i, s_i', a_i') \in \mathcal{D} \,\Big|\, \|s_i - s\|_2^2 \leq d^2 \text{ and } \|s_i' - s_i\|_2^2 \leq \bar{d}^2 \right\} \triangleq \Omega_s^{(d,\bar{d})}. \quad (1)$$

For any transition $(\bar{s}, \bar{a}, \bar{r}, \bar{s}', \bar{a}') \in \Omega_s^{(d,\bar{d})}$, there exists an optimal uniform local weight vector $w_s^*$ such that the local $Q$-function approximation is defined as

$$\hat{Q}_{\Omega_s^{(d,\bar{d})}}(\bar{s}, \bar{a}) \triangleq w_s^{*T} \phi(\bar{s}, \bar{a}), \quad \forall (\bar{s}, \bar{a}, \bar{r}, \bar{s}', \bar{a}') \in \Omega_s^{(d,\bar{d})}, \quad (2)$$

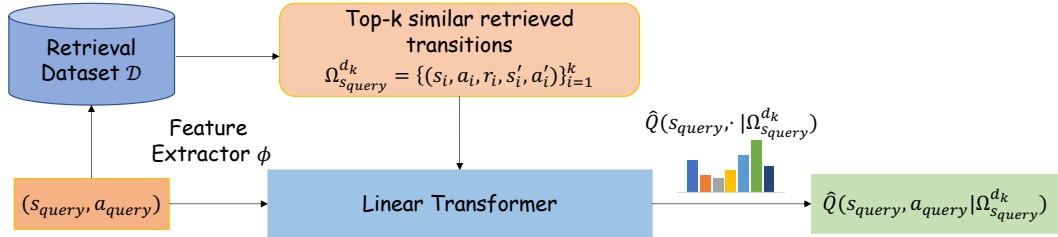

Figure 2: An overview of In-Context Compositional Q-Learning (`ICQL`). Given a query state-action pair $(s_{\text{query}}, a_{\text{query}})$, the model embeds it with the feature extractor $\phi$, retrieves the top-$k$ most similar transitions from the offline dataset $\mathcal{D}$, and forms a local context set. A local linear Q-function $\hat{Q}(s, a | \Omega^{d_k}_{s_{\text{query}}})$ is then estimated from the retrieved context and used to update the actor.

where the function $\phi : \mathcal{S} \times \mathcal{A} \to \mathbb{R}^d$ is the feature function of the state-action pair $(\bar{s}, \bar{a})$. The best approximation to the local Q-function $Q_{\Omega_s^{(d,\bar{d})}}(\bar{s}, \bar{a})$ is $\hat{Q}_{\Omega_s^{(d,\bar{d})}}(\bar{s}, \bar{a})$, that is, there exists some $\varepsilon^s_{\text{approx}} > 0$ such that

$$\left| Q_{\Omega_s^{(d,\bar{d})}}(\bar{s}, \bar{a}) - w^{*\top}_s \phi(\bar{s}, \bar{a}) \right| \le \varepsilon^s_{\text{approx}}, \quad \forall (\bar{s}, \bar{a}, \bar{r}, \bar{s}', \bar{a}') \in \Omega_s^{(d,\bar{d})}. \tag{3}$$

In the remainder of this paper, we omit $\bar{d}$ from the notation of $\Omega_s^{(d,\bar{d})}$ in Equation (1), because the condition $\|\bar{s}' - \bar{s}\|_2^2 \le \bar{d}^2$ for some $\bar{d} > 0$ can be readily satisfied in continuous domains. We use $\Omega_s^d$ to denote $\Omega_s^{(d,\bar{d})}$ instead. The local $Q$-function defined in Equation (2) can be viewed as a localized formulation of general linear $Q$-function approximation, which has been widely used in prior work (Yin et al., 2022; Du et al., 2020; Poupart et al., 2002; Parr et al., 2008). We assume that each local domain $\Omega_s^d$ admits its own state-dependent local structure. This perspective has been studied both theoretically and empirically and has been shown to yield improved $Q$-function approximation and strong performance on complex tasks; see Section B for additional discussion. In practice, the radius $d$ is not directly tunable: it depends on the underlying density and geometry of the dataset and is unknown to the algorithm. Therefore, we adopt a retrieval mechanism with size parameter $k$ to control locality in practice.

## 3.2 RETRIEVAL METHODS

In this section, we introduce the methods used to retrieve transitions from the offline dataset $\mathcal{D}$. We focus on three strategies: state-similar retrieval, random retrieval, and state-similar-with-high-reward retrieval. Each retrieval strategy provides a different level of coverage of the local neighborhood $\Omega^{d_k}_{s_{\text{query}}}$ associated with the query state $s_{\text{query}}$. Both state-similar retrieval and state-similar-with-high-reward retrieval are expected to capture more accurate and comprehensive local information from the local neighborhood $\Omega_s^d$. The difference is that state-similar retrieval can preserve greater diversity in the action space, whereas state-similar-with-high-reward retrieval is intended to recover higher-quality transitions. We define state-similar retrieval in this section. See Section C for additional details and the definitions of the other two retrieval methods.

**Definition 3.2** (State-Similar Retrieval). Given the query state $s_{\text{query}}$, `ICQL` retrieves $k$ transitions with the smallest $l_2$-distance between the retrieved state $s_i$ and $s_{\text{query}}$, *i.e.*,

$$\overline{\Omega}_{s_{\text{query}}} \triangleq \left\{ (s_i, a_i, r_i, s_i', a_i') \in \mathcal{D} \middle| s_i \in \arg \text{top-k} \left\{ -\|s_{\text{query}} - s_i\|_2^2 \right\} \right\}. \tag{4}$$

Let us define $d_k^{s_{\text{query}}} \triangleq \max_{(s_i, a_i, r_i, s_i', a_i') \in \overline{\Omega}_{s_{\text{query}}}} \{\|s_{\text{query}} - s_i\|_2\}$. Then, we have $\overline{\Omega}^k_{s_{\text{query}}} = \Omega^{d_k^{s_{\text{query}}}}_{s_{\text{query}}}$. The quantity $d_k^{s_{\text{query}}}$ depends on the query state $s_{\text{query}}$, but for ease of presentation, we use $d_k$ to denote $d_k^{s_{\text{query}}}$. Because the main version of `ICQL` uses the fixed state-similar retrieval method, we use $\Omega^{d_k}_{s_{\text{query}}}$ to denote the retrieved context provided to `ICQL` for notational consistency. In the next section, we explain how the transitions in $\Omega^{d_k}_{s_{\text{query}}}$ are used to learn the best local Q-function approximation $\hat{Q}_{\Omega^{d_k}_{s_{\text{query}}}}(s, a)$ for all $(s, a, r, s', a') \in \Omega^{d_k}_{s_{\text{query}}}$ through in-context learning.

## 3.3 IN-CONTEXT COMPOSITIONAL Q-LEARNING

We now describe how compositional Q-functions are learned through contextual inference. We first define a context-dependent weight function for estimating the optimal local weight vector $w_s^*$ defined in Definition 3.1 for each state $s$.

**Definition 3.3** (Context-dependent Weights). The local weight function $w_s : \mathcal{P}(\Omega) \to \mathbb{R}^d$ is a context-dependent function inferred through in-context learning or retrieval-based adaptation, where $\mathcal{P}(\Omega) = \{A | A \subseteq \Omega\}$ is the power set of $\Omega$ and $\Omega$ contains all possible transitions for a given task.

We emphasize that the offline dataset $\mathcal{D} \subseteq \Omega$. Based on Definition 3.3, there exists some $\Omega_s^* \subseteq \Omega$ such that $w_s(\Omega_s^*) = w_s^*$. It is not necessary that $\Omega_s^* \subseteq \mathcal{D}$. Different retrieval methods can be used to cover $\Omega_s^*$ as much as possible, thereby improving weight approximation. Then, for any query state $s_{\text{query}}$ and action $a_{\text{query}}$, suppose $\Omega_{s_{\text{query}}}^{d_k}$ is the set of size $N$ containing the $k$ retrieved transitions from $\mathcal{D}$ obtained by the state-similarity criterion defined in Section 3.2. Inspired by Wang et al. (2025b), the input "prompt" matrix for linear Transformer is constructed as

$$Z_0 = \begin{bmatrix} \phi_0 & \cdots & \phi_{N-1} & \phi_{\text{query}} \\ \gamma\phi_0' & \cdots & \gamma\phi_{N-1}' & 0 \\ r_0 & \cdots & r_{N-1} & 0 \end{bmatrix}, \tag{5}$$

where $\phi$ is the state-action-pair feature extractor, $\phi_i \triangleq \phi(s_i, a_i)$ and $\phi_i' \triangleq \phi(s_i', a_i')$ and $\phi_{\text{query}} \triangleq \phi(s_{\text{query}}, a_{\text{query}})$ for any $a_{\text{query}} \in \mathcal{A}$. Assume an $L$-layer linear Transformer, for $\ell = 0, 1, \cdots, L-1$, each layer $\ell$ has weight matrices $P_\ell$ and $G_\ell$ defined as

$$P_\ell \triangleq \begin{bmatrix} 0_{2d \times 2d} & 0_{2d \times 1} \\ 0_{1 \times 2d} & 1 \end{bmatrix}, G_\ell \triangleq \begin{bmatrix} -C_\ell^T & C_\ell^T & 0_{d \times 1} \\ 0_{d \times d} & 0_{d \times d} & 0_{d \times 1} \\ 0_{1 \times d} & 0_{1 \times d} & 0 \end{bmatrix}, \tag{6}$$

where all the matrices $\{C_\ell\}_{\ell=0}^{L-1} \in \mathbb{R}^{d \times d}$ are trainable parameters and $d$ is the dimension of hidden feature. By feeding $Z_0$ through the linear Transformer $TF_\theta^Q$ with linear attention layers $LinAttn(Z; P, G) \triangleq PZM(Z^\top GZ)$ and take the bottom-right element on index $[2d+1, k+1]$, we obtain a context-dependent Q-function approximation denoted as

$$\hat{Q}(s_{\text{query}}, a_{\text{query}} | \Omega_{s_{\text{query}}}^{d_k}) = w_{s_{\text{query}}}^L (\Omega_{s_{\text{query}}}^{d_k})^T \phi(s_{\text{query}}, a_{\text{query}}), \tag{7}$$

which approximates $\hat{Q}_{\Omega_{s_{\text{query}}}^{d_k}}(s, a)$ defined in Equation (2). Each linear attention layer updates $w_{s_{\text{query}}}^L(\Omega_{s_{\text{query}}}^{d_k})$ iteratively for each retrieved transition $(s, a, r, s', a') \in \Omega_{s_{\text{query}}}^{d_k}$:

$$
\begin{aligned}
&w_{s_{\text{query}}}^{l+1}(\Omega_{s_{\text{query}}}^{d_k}) \\
=&w_{s_{\text{query}}}^l(\Omega_{s_{\text{query}}}^{d_k}) + \alpha\Big(r + \gamma\hat{Q}(s', a'|\Omega_{s_{\text{query}}}^{d_k}) - \hat{Q}(s, a|\Omega_{s_{\text{query}}}^{d_k})\Big)\nabla_w\hat{Q}(s, a|\Omega_{s_{\text{query}}}^{d_k}) \\
=&w_{s_{\text{query}}}^l(\Omega_{s_{\text{query}}}^{d_k}) + \alpha\Big(r + \gamma w_{s_{\text{query}}}(\Omega_{s_{\text{query}}}^{d_k})^T\phi(s', a') - w_{s_{\text{query}}}^l(\Omega_{s_{\text{query}}}^{d_k})^T\phi(s, a)\Big)\phi(s, a),
\end{aligned}
\tag{8}
$$

where $\alpha$ is the learning rate, $l$ denotes the index of linear attention layer, the first equality follows from SARSA (Sutton & Barto, 2018), and the second equality follows from Equation (7). See Section D for details on the theorem showing that the proposed ICQL can implement in-context TD learning.

For training ICQL, we follow IQL (Kostrikov et al., 2022) by performing value iteration via expectile regression and policy extraction via advantage-weighted regression. Specifically, the critic loss is defined using our local Q-function approximation:

$$\mathcal{L}_{\text{critic}} = \mathbb{E}_{(s,a,r,s') \sim \mathcal{D}}\left[\rho_\tau\left(\hat{Q}(s, a|\Omega_s^{d_k}) - y\right)\right], \tag{9}$$

where $y = r + \gamma V(s'|\Omega_{s'}^{d_k}), V(s'|\Omega_{s'}^{d_k}) = \mathbb{E}_{a' \sim \pi}\left[\hat{Q}(s', a'|\Omega_{s'}^{d_k})\right]$, $V$ is also a context-dependent value estimator, and $\rho_\tau(\cdot)$ denotes the expectile regression error. The policy is optimized via advantage-weighted regression, using an advantage defined by local value estimation conditioned on the current state and its retrieved similar states:

$$\mathcal{L}_{\text{policy}} = \mathbb{E}_{s \sim \mathcal{D}}\left[\mathbb{E}_{a \sim \pi}\left[\exp\left(\beta \cdot (\hat{Q}(s, a|\Omega_s^{d_k}) - V(s|\Omega_s^{d_k}))\right)\log\pi(a|s)\right]\right]. \tag{10}$$

After training, the extracted policy can be evaluated independently without any additional retrieval or contextual inference. The pseudocode of training is provided in Algorithm 1.

## 3.4 THEORETICAL ANALYSIS ON ICQL

In this section, we analyze the theoretical properties of our algorithm ICQL. ICQL captures the compositional and local structures of complex decision-making tasks by enabling the Q-function to vary flexibly across different state regions. However, the performance of such local approximators depends critically on two factors:

(i) the expressiveness of the feature representation $\phi(s, a)$,

(ii) the accuracy of the learned weight function $w_s(\Omega_s^{d_k})$ in approximating the optimal local weight $w_s^*$ corresponding to the state $s$ and the retrieved offline transition set $\Omega_s^{d_k}$.

To show that the greedy policy induced by ICQL is near-optimal, we first derive pointwise and expected bounds on the local Q-function approximation error, highlighting how both approximation error and weight estimation error contribute to the total error. Building on these results, we further characterize how the approximation error propagates to policy sub-optimality through the performance difference lemma. These analyses provide theoretical justification for the importance of accurate local value estimation in achieving strong policy performance in offline RL settings. In this section, we present only the assumptions required for the analysis and the main near-optimality theorem for ICQL. See Section E for more detailed and comprehensive proofs.

**Assumption 3.1.** Let $\phi : \mathcal{S} \times \mathcal{A} \to \mathbb{R}^d$ be a fixed feature map. We assume that for all $(s, a) \in \mathcal{S} \times \mathcal{A}$, the feature norm is bounded as $\|\phi(s, a)\| \leq B_\phi$.

*Remark* 3.2. Assumption 3.1 is commonly adopted in prior work (Wang & Zou, 2020; Bhandari et al., 2018; Shen et al., 2023). In our experiments, we use a tanh activation function in the last layer of our feature extractor $\phi$, which implies that each component of the feature vector $\phi(s, a)$ is bounded by 1. Hence, we can conclude that $\|\phi(s, a)\| \leq d$, where $d$ is the feature dimension. This remark supports Assumption 3.1.

**Assumption 3.3** (Set Coverage). For each query state $s_{\text{query}} \in \mathcal{S}$, let $\Omega_{s_{\text{query}}}^*$ denote the ideal local transition set defined in Section 3.3. Suppose the retrieved set $\Omega_{s_{\text{query}}}^{d_k}$ satisfies

$$\kappa_{s_{\text{query}}} \triangleq \frac{\left|\Omega_{s_{\text{query}}}^{d_k} \cap \Omega_{s_{\text{query}}}^*\right|}{\left|\Omega_{s_{\text{query}}}^*\right|} \geq \sigma, \tag{11}$$

for some coverage ratio $\sigma \in (0, 1]$. Equivalently, at least $m = \sigma|\Omega_{s_{\text{query}}}^*|$ transitions from $\Omega_{s_{\text{query}}}^*$ are contained in $\Omega_{s_{\text{query}}}^{d_k}$.

*Remark* 3.4. Assumption 3.3 quantifies how many transitions from $\Omega_{s_{\text{query}}}^*$ are covered by the retrieved set $\Omega_{s_{\text{query}}}^{d_k}$. This type of coverage condition is standard in nonparametric regression (Györfi et al., 2002; Devroye et al., 1996; Cover & Hart, 1967; Kpotufe, 2011) and has also been widely adopted in the analysis of offline RL through concentrability or coverage coefficients (Munos, 2003; 2007; Antos et al., 2008; Chen et al., 2019; Xie et al., 2021). The value of $d_k$ and the choice of retrieval method both affect $\kappa_s$. We present an ablation study on the number of retrieved transitions and the retrieval method in Section 4.3.

We now present our main theorem, which shows that the performance of the greedy policy with respect to the learned local Q-function approximation $\hat{Q}(s, a|\Omega_{s_{\text{query}}}^{d_k})$ is near-optimal.

**Theorem 3.5** (Policy Performance Gap). *Suppose Assumptions 3.1 and 3.3 hold, and the learned policy $\pi$ is greedy with respect to $\hat{Q}(s, a|\Omega_s^{d_k})$. Then, with probability at least $1 - \delta$, the performance gap is bounded as*

$$J(\pi^*) - J(\pi) \leq \frac{2}{1 - \gamma} \mathbb{E}_{s \sim d^\pi} \left[ \varepsilon_{approx}^s(1 + B_\phi) + CB_\phi\sqrt{\frac{d + \log(1/\delta)}{\sigma\,|\Omega_s^{d_k}|}} \right], \tag{12}$$

*where $C > 0$ depends on $B_\phi$, $B_r$ and the conditioning of the local Gram matrix.*

*Proof.* See Section E.1 for details. □

# 4 EXPERIMENTS

In this section, we evaluate `ICQL` on standard offline RL benchmarks and analyze its empirical behavior. We first describe the benchmark environments and datasets used in our experiments. Then we present the results and analyze the performance of `ICQL` across tasks and variants.

## 4.1 ENVIRONMENTS AND DATASETS

We evaluate our method on a diverse set of continuous control benchmarks from the D4RL suite (Fu et al., 2020), which includes three types of offline reinforcement learning environments:

**MuJoCo tasks** (e.g., `HalfCheetah-Medium`) are standard locomotion environments based on MuJoCo (Todorov et al., 2012), featuring smooth dynamics and dense rewards. These tasks are commonly used to assess sample efficiency and stability.

**Adroit tasks** (e.g., `Pen-Human`) involve high-dimensional dexterous manipulation using a 24-DoF robotic hand. The action spaces are complex and the datasets are collected from human demonstration or behavior imitation, making them challenging due to limited action coverage.

**Kitchen tasks** (e.g., `Kitchen-Complete`) are long-horizon goal-conditioned tasks that require solving compositional subtasks (e.g., turning on lights, opening cabinets). These tasks emphasize multi-stage behavior and compositional reasoning.

## 4.2 MAIN RESULTS

We compare our method against five widely adopted offline RL algorithms: `BC`, `DT` (Chen et al., 2021), `TD3+BC` (Fujimoto & Gu, 2021), `CQL` (Kumar et al., 2020) and `IQL` (Kostrikov et al., 2022). These baselines represent two complementary paradigms: the first three represent policy constraints, and the last two represent value regularization. The experiment results are shown in Table 1.

Table 1: Performance comparison across Mujoco, Adroit, and Kitchen tasks. Average and standard deviation of scores are reported over 5 random seeds.

| Mujoco Tasks | BC | DT | TD3+BC | CQL | IQL | ICQL(Ours) | Gain(%) |
|---|---|---|---|---|---|---|---|
| Walker2d-Medium-Expert-v2 | 107.5 | 70.7 | 109.2 | 98.7 | 109.8 | $\mathbf{113.3}_{\pm 2.0}$ | 3.1% |
| Walker2d-Medium-v2 | 75.3 | 70.2 | 77.0 | 79.2 | 71.5 | $\mathbf{80.3}_{\pm 5.2}$ | 1.4% |
| Walker2d-Medium-Replay-v2 | 26.0 | 54.8 | 41.5 | 77.2 | 61.0 | $\mathbf{81.9}_{\pm 5.4}$ | 6.1% |
| Hopper-Medium-Expert-v2 | 52.5 | 57.5 | 78.2 | 105.4 | 98.5 | $\mathbf{111.0}_{\pm 0.6}$ | 5.3% |
| Hopper-Medium-v2 | 52.9 | 57.1 | 53.5 | 58.0 | 63.3 | $62.6_{\pm 7.9}$ | -1.5% |
| Hopper-Medium-Replay-v2 | 18.1 | 65.8 | 59.4 | 95.0 | 82.4 | $\mathbf{96.4}_{\pm 4.9}$ | 1.5% |
| HalfCheetah-Medium-Expert-v2 | 55.2 | 70.8 | 62.8 | 62.4 | 83.4 | $\mathbf{89.1}_{\pm 4.2}$ | 6.8% |
| HalfCheetah-Medium-v2 | 42.6 | 42.8 | 43.1 | 44.4 | 42.5 | $\mathbf{45.9}_{\pm 0.3}$ | 3.5% |
| HalfCheetah-Medium-Replay-v2 | 36.6 | 39.5 | 41.8 | **45.5** | 38.9 | $44.7_{\pm 0.1}$ | -1.8% |
| **Average** | 51.9 | 58.8 | 62.9 | 74.0 | 72.4 | **80.6** | 8.8% |
| **Adroit Tasks** | **BC** | **DT** | **TD3+BC** | **CQL** | **IQL** | **ICQL** | **Gain(%)** |
| Pen-Human-v1 | 63.9 | 79.5 | 64.6 | 37.5 | **89.5** | $85.6_{\pm 5.6}$ | -4.3% |
| Pen-Cloned-v1 | 37.0 | 74.0 | 76.8 | 39.2 | 4.9 | $\mathbf{89.4}_{\pm 4.8}$ | 5.4% |
| Hammer-Human-v1 | 1.2 | 1.7 | 1.5 | 4.4 | **7.2** | $3.7_{\pm 3.2}$ | -49.4% |
| Hammer-Cloned-v1 | 0.6 | 3.7 | 1.8 | 2.1 | 0.5 | $\mathbf{4.5}_{\pm 5.5}$ | 23.4% |
| Door-Human-v1 | 2.0 | 5.5 | 0.2 | 9.9 | 9.8 | $\mathbf{17.1}_{\pm 5.5}$ | 73.1% |
| Door-Cloned-v1 | 0.0 | 3.2 | -0.1 | 0.1 | 7.6 | $\mathbf{11.7}_{\pm 4.4}$ | 53.6% |
| **Average** | 17.45 | 27.9 | 24.2 | 15.5 | 33.2 | **35.3** | 6.3% |
| **Kitchen Tasks** | **BC** | **DT** | **TD3+BC** | **CQL** | **IQL** | **ICQL** | **Gain(%)** |
| Kitchen-Complete-v0 | 65.0 | 52.5 | 57.5 | 43.8 | 59.2 | $\mathbf{79.3}_{\pm 2.1}$ | 22.0% |
| Kitchen-Mixed-v0 | 51.5 | **60.0** | 53.5 | 51.0 | 53.3 | $59.5_{\pm 6.0}$ | -0.8% |
| Kitchen-Partial-v0 | 38.0 | 55.0 | 46.7 | 49.8 | 45.8 | $\mathbf{61.5}_{\pm 5.8}$ | 11.8% |
| **Average** | 51.5 | 55.8 | 52.6 | 48.2 | 52.8 | **66.8** | 16.4% |

Results demonstrate that, on MuJoCo tasks, `ICQL` outperforms second best baseline `CQL` by 8.8% on average. On Adroit tasks, `ICQL` improves `IQL` by 6.3%. Notably, on Kitchen task, `ICQL` achieves a **16.4% improvement** over `DT` on Kitchen tasks, highlighting the importance of compositional value estimation in environments with complex, multi-stage structure. However on Hammer-Human dataset, `ICQL` is inferior to two baseline methods, which may relate to the dataset quality issue. In `Hammer-Human`, the size of the dataset is smaller and the distance between query states and retrieved similar states are larger than those of `Hammer-Cloned`, making it harder for in-context learning. The detailed analysis on this issue is provided in Section H.4. Although `ICQL` improves performance through in-context learning, the additional computational overhead remains moderate; a detailed analysis is provided in Section H.3. Overall, these results validate the general applicability of `ICQL` across both value-learning and actor-critic paradigms.

For investigating whether `ICQL` can produce more accurate value estimation than baseline methods, we conduct analysis on the learned Q function by comparing the Q prediction among `ICQL`, `IQL` and online RL method `SAC`. We plot their Q estimations of the same set of offline dataset entries, and leverage t-SNE for showing their respective Q-estimate distribution over the same state space. Figure 3 shows the results on `Walker2d-Medium` dataset, where `ICQL` shares similarity score of 0.69 with `SAC` on Q estimation, while `IQL` can only achieve a similarity score about 0.29. This indicates that the superior performance of `ICQL` on `IQL` comes from a better Q estimation, ensured by local Q function estimation, over the noisy dataset.

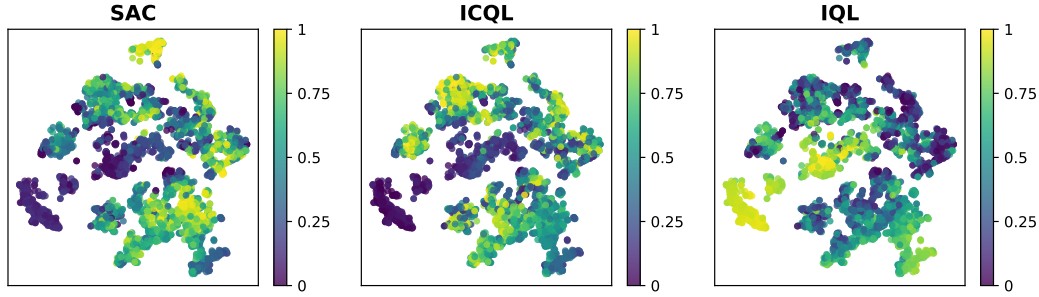

Figure 3: Q-value distribution on states after t-SNE dimension reduction, of `Walker2d-Medium` dataset. The partitioned value patterns support our hypothesis that Q-functions are inherently compositional, motivating localized value modeling.

### 4.3 ABLATION STUDIES

In this section, we study how different design choices affect the performance of `ICQL`, including the number of in-context learning layers, the context length, retrieval strategy and in-context learning module architecture.

#### 4.3.1 NUMBER OF IN-CONTEXT LEARNING LAYERS

In this experiment, we investigate the effect of in-context learning steps, which is controlled by the number of layers in the in-context critic network. The number of layers is selected from $\{4, 8, 16, 20\}$. The experiments are conducted on MuJoCo tasks. Figure 4 displays the experiment outcomes and detailed numerical results are provided in Appendix Table 8. From Figure 4, the normalized scores generally increase as the number of layers increase in most of the tasks, indicating that a larger number of layers may lead to more sufficient in-context value-learning. While the phenomenon is not obvious in Hopper tasks, one possible reason is the significant distribution shift in Hopper environment due to the high variance of transition dynamics.

#### 4.3.2 INFLUENCE OF CONTEXT LENGTH

In this experiment, we investigate the effect of context lengths in `ICQL`. The context lengths are selected from $\{10, 20, 30, 40\}$. As shown in Figure 5, a context length of 20 yields the generally best performance for in-context TD-learning in Gym tasks, where too long or too short context lengths

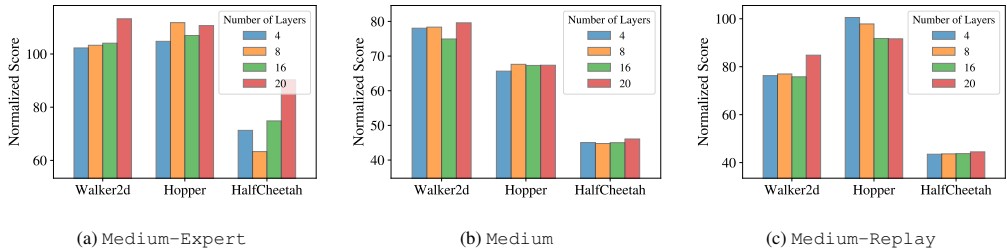

Figure 4: Normalized scores of different number of in-context learning layers on Mujoco tasks. Each color represents different number of layers, and the y-axis represents the normalized score.

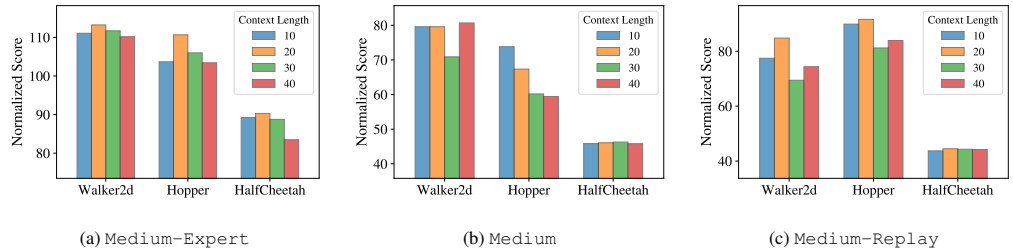

Figure 5: Normalized scores of context lengths on Mujoco tasks. Each color represents different context lengths, and the y-axis represents the normalized score.

lead to sub-optimal results. These results provide evidence that the "locality" of context is crucial for in-context learning performance. As the context length increases, the distance between query state and context transitions also gets larger, which may break the "local" definition and bring noise into the in-context learning process. Detailed numerical results are shown in Table 8.

### 4.3.3 CONTEXT RETRIEVAL STRATEGIES

In this experiment, we investigate the impact of retrieval quality by applying different context retrieval strategies to `ICQL`. In addition to the standard **State-Similar Retrieval**, we compare two additional retrieval strategies: (1) **Random Retrieval**, which selects transitions uniformly at random from the offline dataset; and (2) **State-Similar-with-High-Rewards Retrieval**, which further filters similar-state candidates by selecting those with higher rewards. The definitions of these three retrieval methods are provided in Section 3.2 and Section C.

Table 2: Ablation study on retrieval strategies of `ICQL`. We compare three variants: **Random Retrieval**, **State-Similar Retrieval**, and **State-Similar-with-High-Rewards Retrieval**.

| Dataset | Random | State-Similar | State-Similar-with-High-Rewards |
|---|---|---|---|
| Walker2d-Medium-v2 | 78.1 | 80.3 | **83.9** |
| Walker2d-Medium-Replay-v2 | 67.5 | **81.9** | 75.1 |
| Hopper-Medium-v2 | **74.1** | 62.6 | 59.9 |
| Hopper-Medium-Replay-v2 | 81.0 | **96.4** | 90.8 |
| HalfCheetah-Medium-v2 | 45.5 | 45.9 | **46.4** |
| HalfCheetah-Medium-Replay-v2 | 43.4 | **44.7** | 43.2 |
| Pen-Human-v1 | 75.1 | **85.6** | 84.8 |
| Hammer-Human-v1 | 1.4 | 3.7 | **4.4** |
| Door-Human-v1 | 12.0 | **17.1** | 15.6 |
| Kitchen-Complete-v0 | 70.0 | **79.3** | 71.3 |
| Kitchen-Mixed-v0 | 53.8 | 59.5 | **60.0** |
| Kitchen-Partial-v0 | 47.5 | **61.5** | 50.0 |

Our results show that **State-Similar Retrieval** provides the strongest overall performance and is the best-performing strategy on most tasks, demonstrating the benefit of constructing context from locally

similar states. **Random Retrieval** is generally less effective and often yields inferior performance, which highlights the importance of context relevance in local value estimation. Meanwhile, **State-Similar-with-High-Rewards Retrieval** outperforms the other strategies on several tasks, including `walker2d-medium`, `halfcheetah-medium`, `hammer-human`, and `kitchen-mixed`. This suggests that incorporating reward information into retrieval can be beneficial when high-value transitions are especially informative, while pure state similarity remains the most robust choice overall.

### 4.3.4 COMPARISON ACROSS DIFFERENT IN-CONTEXT MODELING CHOICES

We compare different architectural choices for the in-context module in `ICQL`, including a linear Transformer, a small linear MLP, and a standard self-attention-based Transformer. The results are shown in Table 3. Overall, the linear Transformer achieves the best performance on the large majority of tasks and shows especially clear advantages on replay-style and long-horizon datasets, such as `Walker2d-Medium-Replay`, `Hopper-Medium-Replay`, and the Kitchen benchmarks. The standard Transformer is generally less stable and performs substantially worse on several tasks, while the linear MLP is more competitive but still underperforms the linear Transformer in most cases. Although the linear MLP performs best on `Hammer-Human` and `Hammer-Cloned`, this advantage does not transfer consistently to other domains. These results suggest that the proposed linear in-context mechanism is not only theoretically convenient, but also empirically important for stable and effective local Q-function estimation across diverse offline RL environments.

Table 3: Performance comparison across different local modeling choices: linear attention-based Transformer, linear MLP, and standard self-attention-based Transformer.

| Task | Linear Transformer | Linear MLP | Standard Transformer |
|---|---|---|---|
| Walker2d-Medium-Expert | **113.3** | 109.5 | 108.8 |
| Walker2d-Medium | **80.3** | 76.7 | 77.4 |
| Walker2d-Medium-Replay | **81.9** | 60.2 | 42.9 |
| Hopper-Medium-Expert | **111.0** | 109.9 | 70.3 |
| Hopper-Medium | **62.6** | 55.7 | 61.9 |
| Hopper-Medium-Replay | **96.4** | 89.9 | 42.1 |
| HalfCheetah-Medium-Expert | **89.1** | 83.0 | 72.5 |
| HalfCheetah-Medium | **45.9** | 43.3 | 42.0 |
| HalfCheetah-Medium-Replay | **44.7** | 39.2 | 36.1 |
| Pen-Human | **85.6** | 66.6 | 72.7 |
| Pen-Cloned | **89.4** | 80.7 | 83.8 |
| Hammer-Human | 3.7 | **6.1** | 4.2 |
| Hammer-Cloned | 4.5 | **7.9** | 1.8 |
| Door-Human | **17.1** | 6.9 | 8.9 |
| Door-Cloned | **11.7** | 3.5 | 3.4 |
| Kitchen-Complete | **79.3** | 70.0 | 78.3 |
| Kitchen-Mixed | **59.5** | 57.5 | 55.8 |
| Kitchen-Partial | **61.5** | 48.3 | 55.8 |

## 5 CONCLUSION AND FUTURE WORK

We introduced `ICQL`, a novel offline RL framework that casts value estimation as an in-context inference problem using linear attention. By retrieving local transitions and fitting context-dependent local Q-functions, `ICQL` enables compositional reasoning without requiring subtask supervision. We provide theoretical guarantees showing that greedy action extraction based on `ICQL` yields a near-optimal policy. Experiments show that `ICQL` achieves strong performance gains and produces value estimates that are closer to those of online RL algorithms. These results highlight the potential of in-context learning as a powerful inductive bias for offline reinforcement learning. Although the methodology of `ICQL` is agnostic to the choice of distance metric, retrieval quality remains a practical concern in complex, high-dimensional state spaces. An important and promising direction for future work is to combine `ICQL` with more sophisticated retrieval methods, such as pre-trained state encoders or value-aware learnable retrievers.

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

## LLM USAGE STATEMENT

LLMs were used to aid the writing and polishing of the manuscript.

## APPENDIX

## A  PRELIMINARY

### A.1  REINFORCEMENT LEARNING

Consider an infinite-horizon Markov Decision Process (MDP) defined by the tuple $(\mathcal{S}, \mathcal{A}, p_0, p_{\text{MDP}}, \mathcal{R}, \gamma)$, where $\mathcal{S}$ and $\mathcal{A}$ denote the state and action spaces, respectively. The reward function is $\mathcal{R} : \mathcal{S} \times \mathcal{A} \to \mathbb{R}$, and the transition dynamics are governed by $p_{\text{MDP}}(s'|s, a)$, which denotes the probability of transitioning to state $s'$ from state $s$ after taking action $a$. The initial state distribution is $p_0 : \mathcal{S} \to [0, 1]$, and $\gamma \in [0, 1)$ is the discount factor.

At each time step $t$, the agent observes state $s_t$, selects an action $a_t \sim \pi(\cdot|s_t)$ according to a stochastic policy $\pi : \mathcal{A} \times \mathcal{S} \to [0, 1]$, receives a reward $r_t = \mathcal{R}(s_t, a_t)$, and transitions to the next state $s_{t+1} \sim p_{\text{MDP}}(\cdot|s_t, a_t)$. This interaction generates trajectories of the form $(s_0, a_0, r_0, s_1, a_1, r_1, \dots)$.

Given a policy $\pi$, the associated Q-function and value function quantify the expected cumulative discounted rewards starting from the state-action pair $(s_t, a_t)$ and the state $s_t$, respectively:

$$Q^\pi(s_t, a_t) \triangleq \mathbb{E}_{a_{t+1}, a_{t+2}, \cdots \sim \pi} \left[ \sum_{i=0}^{\infty} \gamma^i \mathcal{R}(s_{t+i+1}, a_{t+i+1})|s_t, a_t \right], \tag{13a}$$

$$V^\pi(s_t) \triangleq \mathbb{E}_{a_t \sim \pi(\cdot|s_t)} \left[ Q^\pi(s_t, a_t) \right]. \tag{13b}$$

The Q-function satisfies the *Bellman Expectation Equation*:

$$Q^\pi(s,a) = \mathcal{R}(s,a) + \gamma \, \mathbb{E}_{s' \sim p_{\text{MDP}}(\cdot|s,a)} \left[ V^\pi(s') \right]. \tag{14}$$

Similarly, the value function satisfies:

$$V^\pi(s) = \mathbb{E}_{a \sim \pi(\cdot|s)} \left[ Q^\pi(s,a) \right]. \tag{15}$$

The goal of reinforcement learning is to learn a policy $\pi_\theta(a|s)$ that maximizes the expected cumulative discounted reward. The optimal value functions satisfy the *Bellman Optimality Equations*:

$$Q^*(s,a) = \mathcal{R}(s,a) + \gamma \, \mathbb{E}_{s' \sim p_{\text{MDP}}(\cdot|s,a)} \left[ \max_{a'} Q^*(s',a') \right], \tag{16a}$$

$$V^*(s) = \max_{a \in \mathcal{A}} Q^*(s,a). \tag{16b}$$

In the offline setting, rather than interacting with the environment, the agent is given a fixed dataset $\mathcal{D} = \{(s,a,r,s')\}$ collected by a behavior policy $\pi_\beta$. Offline RL algorithms aim to learn an effective policy entirely from this static dataset $\mathcal{D}$, without any further interaction with the environment. A central challenge in offline RL is the *distributional shift* (Kumar et al., 2019; Jaques et al., 2019; Levine et al., 2020; Wu et al., 2019) between the learned policy $\pi$ and the behavior policy $\pi_\beta$, which often leads to overestimation and poor generalization when estimating Q-values for out-of-distribution state-action pairs.

## A.2 IN-CONTEXT LEARNING WITH LINEAR ATTENTIONS

Recently, there has been significant interest in understanding the theoretical capabilities of in-context learning with linear attention mechanisms (Dai et al., 2023; Ahn et al., 2023; 2024), particularly in random instances of linear regression and simple classification tasks. We formally introduce these problem settings in this section. Throughout this paper, all vectors are treated as column vectors. We denote the identity matrix in $\mathbb{R}^n$ by $I_n$, and the $m \times n$ all-zero matrix by $0_{m \times n}$. For any matrix $Z$, we use $Z^\top$ to denote its transpose, and we use both $\langle x, y \rangle$ and $x^\top y$ interchangeably to denote the inner product.

We define a prompt matrix $Z \in \mathbb{R}^{(d+1) \times (n+1)}$ as follows:

$$Z \triangleq \begin{bmatrix} z^{(0)} & z^{(1)} & \dots & z^{(n-1)} & z^{(n)} \end{bmatrix} = \begin{bmatrix} x^{(0)} & x^{(1)} & \dots & x^{(n-1)} & x^{(n)} \\ y^{(0)} & y^{(1)} & \dots & y^{(n-1)} & 0 \end{bmatrix}, \tag{17}$$

where $\{x^{(i)}, y^{(i)}\}_{i=0}^{n-1}$ are context examples, $x^{(n)}$ is the query input with its corresponding response value $y^{(n)}$ masked as zero, and each $x^{(i)} \in \mathbb{R}^d$ and $y^{(i)} \in \mathbb{R}$ for all $i = 0, \cdots, n$. Following (Von Oswald et al., 2023), we define linear self-attention over the same prompt as

$$\text{LinAttn}(Z; P, G) \triangleq PZM \left( Z^\top G Z \right), \tag{18}$$

where $P, G \in \mathbb{R}^{(d+1) \times (d+1)}$ are learnable parameter matrices, and $M \in \mathbb{R}^{(n+1) \times (n+1)}$ is a fixed mask matrix defined as

$$M \triangleq \begin{bmatrix} I_n & 0_{n \times 1} \\ 0_{1 \times n} & 0 \end{bmatrix}. \tag{19}$$

The goal of training linear Transformers in this setting is to recover the unknown response variable corresponding to $x^{(n)}$, which is represented as zero in the prompt matrix $Z$. By appropriately constructing the parameter matrices $P$ and $G$, the linear attention model in Equation (18) can perform in-context learning for linear regression and simple classification tasks. However, the ability of such models to perform in-context learning for offline reinforcement learning remains poorly understood. Moreover, these analyses are purely theoretical and have not been empirically validated on practical tasks. Transformers can perform in-context supervised learning by mimicking gradient descent updates (Von Oswald et al., 2023), and they can perform in-context reinforcement learning through TD-like methods via appropriately constructed linear attention mechanisms (Wang et al., 2025b). However, (Wang et al., 2025b) considers only the simplified setting of Markov Reward Processes (MRPs), where transitions and rewards depend only on the current state, *i.e.*, $s_{t+1} \sim p(\cdot|s_t)$ and $r_{t+1} = r(s_t)$, with time-dependent context representations. More precisely, their formulation assumes

that each trajectory consists only of temporally contiguous steps. These restrictive assumptions do not hold in real-world decision-making problems, and the empirical results in that work are limited to synthetic MRPs, which makes it difficult to assess how well the method would perform on practical RL tasks. To bridge this gap, we extend the analysis from MRPs to the more general MDP setting by directly estimating the state-action value function $Q(s, a)$ and removing time dependence from the context representations.

## B    OTHER RELATED WORK

**Goal-conditioned and Hierarchical RL.** Goal-conditioned methods such as `UVFA` (Schaul et al., 2015) and `HER` (Andrychowicz et al., 2017) condition policies or value functions on explicit goal inputs to facilitate generalization across tasks. Extensions to compositional settings further decompose Q-functions into subgoal components (Arora, 2024). However, these approaches assume access to goal specifications or subtask labels, which are typically unavailable in offline settings. `ICQL` addresses this limitation by learning Q-functions conditioned on retrieved transition contexts, thereby eliminating the need for task supervision and improving sample efficiency. Hierarchical reinforcement learning decomposes tasks into subgoals or options, thereby enabling temporal abstraction and subpolicy reuse. Classical methods such as `MAXQ` (Dietterich, 2000), `Option-Critic` (Bacon et al., 2017), and `HIRO` (Nachum et al., 2018) explicitly model subtask boundaries and learn separate value functions for each subtask. Although effective when task structure is known or can be discovered, these methods often rely on subgoal specification or auxiliary termination conditions. In contrast, `ICQL` operates without predefined subtask structure and efficiently leverages offline data to obtain a provably accurate local value-function approximation. Unsupervised RL methods such as `DIAYN` (Eysenbach et al., 2019) and `SMiRL` (Berseth et al., 2021) aim to discover diverse behaviors or latent subpolicies without external rewards or supervision. Although these methods can implicitly uncover structure, they are typically designed for unsupervised exploration or pretraining rather than for accurate value estimation in offline settings. `ICQL` instead focuses on precise local Q-function inference conditioned on retrieved experiences, thereby improving compositional generalization and training stability in offline RL.

**Linear Q-function Approximation.** Linear $Q$-function approximation has been widely used in prior work (Yin et al., 2022; Du et al., 2020; Poupart et al., 2002; Parr et al., 2008). Metric MDPs (Kakade et al., 2003), which define the Q-function according to a state-distance metric, are a natural complement to more direct parametric assumptions on value functions and dynamics (Kakade et al., 2003). However, this line of work does not consider local linear Q-function approximation based on a state-distance metric. In our work, we focus on learning improved approximations of local value functions, whereas Kakade et al. (2003) studies accurate approximation of the local environment. We assume that each local domain $\Omega_s^d$ admits its own state-dependent local structure. This perspective has been studied both theoretically and empirically and has been shown to yield improved Q-function approximation and strong performance on complex tasks.

## C    DETAILED DEFINITIONS OF RETRIEVAL STRATEGIES

Retrieval-based methods have shown strong performance across a wide range of domains (Wang et al., 2024; 2025a). In Section 4.3, we compare the performance of `ICQL` under different context-retrieval strategies for approximating the localized Q-function. The retrieval strategies discussed in Section 4.3 are described as follows:

- **State-Similar Retrieval**: Given the current state $s$, search for 20 similar states $s_i$ from the offline dataset using cosine similarity, and retrieve the corresponding transitions $\{s_i, a_i, r_i, s_i', a_i'\}$.
- **Random Retrieval**: Given the current state $s$, randomly select 20 transitions $\{s_i, a_i, r_i, s_i', a_i'\}$ as context.
- **State-Similar-with-High-Rewards**: Given the current state $s$, search for 60 similar states $s_i$ from the offline dataset using cosine similarity, and retrieve the corresponding transitions $\{s_i, a_i, r_i, s_i', a_i'\}$. Then sort the retrieved transitions by reward $r_i$ and select the 20 transitions with the highest rewards as context.

We have provided the mathematical definition of state-similar retrieval in Definition 3.2. Here, we further provide the definitions of the other two retrieval methods—random retrieval and state-similar-with-high-rewards retrieval.

**Definition C.1** (Random Retrieval). Given the query state $s_{\text{query}}$, the randomly retrieved context for `ICQL` is defined as

$$\overline{\Omega}_{s_{\text{query}}}^{\text{random}} \triangleq \left\{ (s_i, a_i, r_i, s_i', a_i') \in \mathcal{D} \,\middle|\, (s_i, a_i, r_i, s_i', a_i') \sim \mathcal{D} \right\}_{i=0}^{k-1}. \tag{20}$$

**Definition C.2** (State-Similar-with-High-Rewards Retrieval). Given the query state $s_{\text{query}}$, $\overline{\Omega}_{s_{\text{query}}}^{\text{high}}$ for `ICQL` is defined as the set of $k$ transitions with the smallest $l_2$-distance between the retrieved state $s_i$ and $s_{\text{query}}$ and the highest transition reward $r_i$, i.e.,

$$\overline{\Omega}_{s_{\text{query}}}^{\text{high}} \triangleq \left\{ (s_i, a_i, r_i, s_i', a_i') \in \overline{\Omega}_{s_{\text{query}}}^{k_s} \,\middle|\, (s_i, a_i, r_i, s_i', a_i') \in \arg\text{top-k}\, \{r_i\} \right\}, \tag{21}$$

where $\overline{\Omega}_{s_{\text{query}}}^{k_s}$ is defined in Equation (4).

For the retrieval methods defined in Definitions C.1 to C.2, we can relate them to Equation (1) by letting $d_1 \triangleq \min_{(s_i,a_i,r_i,s_i',a_i')\in\overline{\Omega}_{s_{\text{query}}}^{k}} \left\{ \|s_i - s_{\text{query}}\|_2 \right\}$ and $d_2 \triangleq \min_{(s_i,a_i,r_i,s_i',a_i')\in\overline{\Omega}_{s_{\text{query}}}^{\text{high}}} \left\{ \|s_i - s_{\text{query}}\|_2 \right\}$. Therefore, we can conclude that $\overline{\Omega}_{s_{\text{query}}}^{k} \subseteq \Omega_{s_{\text{query}}}^{d_1}$ and $\overline{\Omega}_{s_{\text{query}}}^{\text{high}} \subseteq \Omega_{s_{\text{query}}}^{d_2}$, which implies that both state-similar retrieval and state-similar-with-high-reward retrieval can be bounded by some local neighborhood associated with the query state $s_{\text{query}}$.

# D DETAILS OF LINEAR TRANSFORMERS IN `ICQL`

In this section, we show the validity that `ICQL` implements the weight update defined in Equation (8). We first introduce the following lemma, which is motivated by the work of (Wang et al., 2025b) on MRPs.

**Lemma D.1.** *Consider the input $Z_0$ and matrix weights $P_0$ and $G_0$, where*

$$Z_0 = \begin{bmatrix} v_0^{(0)} & \cdots & v_0^{(N-1)} & v_0^{(N)} \\ \xi_0^{(0)} & \cdots & \xi_0^{(N-1)} & \xi_0^{(N)} \\ y_0^{(0)} & \cdots & y_0^{(N-1)} & y_0^{(N)} \end{bmatrix}, P_0 \doteq \begin{bmatrix} 0_{2d\times 2d} & 0_{2d\times 1} \\ 0_{1\times 2d} & 1 \end{bmatrix}, G_0 \doteq \begin{bmatrix} -C_0^T & C_0^T & 0_{d\times 1} \\ 0_{d\times d} & 0_{d\times d} & 0_{d\times 1} \\ 0_{1\times d} & 0_{1\times d} & 0 \end{bmatrix},$$

$$\tag{22}$$

*and $v^{(i)}, \xi^{(i)} \in \mathbb{R}^d$, $y^{(i)} \in \mathbb{R}$. Let $Z_1 \triangleq \text{LinAttn}(Z_0; P_0, G_0) = P_0 Z_0 M(Z_0^T G_0 Z_0)$, and let $y_1^{(N)}$ be the bottom-right element of the next layer output, i.e., $y_1^{(N)} \triangleq Z_1[2d+1, N+1]$. Then it holds that $y_1^{(N)} = -\langle \phi_N, w_1 \rangle$, where*

$$w_1 = w_0 + \frac{1}{N} C_0 \sum_{i=0}^{N-1} (y_0^{(i)} + w_0^T \xi_0^{(i)} - w_0^T v_0^{(i)}) v_0^{(i)}. \tag{23}$$

Using the lemma above, we are ready to prove Theorem D.2.

**Theorem D.2.** *Consider the $L$-layer linear Transformer following Equation (18). Let the matrices $\{P_\ell, G_\ell\}_{\ell=0}^{L-1}$, the mask matrix $M$, and the input prompt matrix $Z_0$ be defined in Equations (5), (6), and (19), respectively. Then the bottom-right element of the $\ell$-th layer output, $Z_\ell[2d+1, n+1]$, satisfies $Z_\ell[2d+1, n+1] = -\langle \phi_{query}, w_{s_{\text{query}}}^\ell(\Omega_{s_{\text{query}}}^{d_k}) \rangle$, where $\{w_{s_{\text{query}}}^\ell(\Omega_{s_{\text{query}}}^{d_k})\}$ is defined by $w_{s_{\text{query}}}^0(\Omega_{s_{\text{query}}}^{d_k}) = 0$ and, for $\ell \geq 0$,*

$$w_{s_{\text{query}}}^{\ell+1}(\Omega_{s_{\text{query}}}^{d_k})$$
$$= w_{s_{\text{query}}}^\ell(\Omega_{s_{\text{query}}}^{d_k}) + \frac{1}{N} C_\ell \sum_{j=0}^{N-1} (r_j + \gamma w_{s_{\text{query}}}^\ell(\Omega_{s_{\text{query}}}^{d_k})^T \phi_j' - w_{s_{\text{query}}}^\ell(\Omega_{s_{\text{query}}}^{d_k})^T \phi_j) \phi_j. \tag{24}$$

---

**Algorithm 1** In-context Q-Learning (`ICQL`)

---

1: **Input:** Offline dataset $\mathcal{D}$, the number of retrieved transitions $k$, feature dimension $d$.
2: **Initialize:** Linear transformer $TF_\theta^Q$ with parameters $\theta$, feature extractor $\phi$.
3: Sample a trajectory $\{(s_i, a_i, r_i)\}_{i=0}^{T-1} \sim \mathcal{D}$.
4: For each query state $s_i$, retrieve $k$ states $s_i^0, \cdots, s_i^{k-1}$ using the state-similar retrieval method defined in Definition 3.2 and extract the corresponding transitions $\{(s_i^j, a_i^j, r_i^j, s_i'^j, a_i'^j)\}_{j=0}^{k-1}$.
5: //In-context Q value estimation.
6: **for** $t = 0, \ldots, T-1$ **do**
7:     Construct the input prompt matrix $Z_t$ by Equation (5).
8:     $\hat{Q}_t \leftarrow TF_\theta^Q(Z_t)[2d+1, k+1]$ by Equation (18).
9: **end for**
10: Update the parameters $\theta$, $\phi$ using Equation (9) and Equation (10).

---

*Proof.* Let $v_0^{(i)} = \phi_i = \phi(s_i, a_i)$, $\xi_0^{(i)} = \gamma\phi_i' = \gamma\phi(s_i', a_i')$, $y_0^{(i)} = r_i$ for $i \in \{0, \cdots, N-1\}$ and $v_0^{(N)} = \phi_{\text{query}} = \phi(s_{\text{query}}, a_{\text{query}})$, $\xi_0^{(N)} = 0_{d\times 1}$, $y_0^{(N)} = 0$, we get

$$w_{s_{\text{query}}}^1(\Omega_{s_{\text{query}}}^{d_k}) = w_{s_{\text{query}}}^0(\Omega_{s_{\text{query}}}^{d_k}) + \frac{1}{N}C_0\sum_{i=0}^{N-1}(r_i + \gamma w_{s_{\text{query}}}^0(\Omega_{s_{\text{query}}}^{d_k})^T\phi_i' - w_{s_{\text{query}}}^0(\Omega_{s_{\text{query}}}^{d_k})^T\phi_i)\phi_i,$$

which is the update rule for pre-conditioned SARSA. We also have

$$y_1^{(N)} = -\langle w_{s_{\text{query}}}^1(\Omega_{s_{\text{query}}}^{d_k}), \phi_{\text{query}}\rangle.$$

By induction on the number of layers $\ell$, the proof is complete. $\square$

## E   PROOFS

In this section, we first derive pointwise and expected bounds on the Q-function approximation error, highlighting how both approximation error and weight-estimation error contribute to the total error. Building on these results, we then characterize how approximation error propagates to policy suboptimality through the performance difference lemma. These analyses provide theoretical justification for the importance of accurate local value estimation in achieving strong policy performance, particularly in offline RL settings.

**Theorem E.1** (Weight Error under Coverage). *Suppose Assumption 3.3 holds, and that the feature vectors are bounded as $\|\phi(s, a)\| \leq B_\phi$ and the rewards are bounded as $|r| \leq B_r$. Let $w_s^*$ be the optimal local weight vector defined in Definition 3.1, and let $w_s(\Omega_s^{d_k})$ be the weight estimated from the retrieved set. Then, with probability at least $1 - \delta$, the following holds:*

$$\left\|w_s(\Omega_s^{d_k}) - w_s^*\right\| \leq C\left(\sqrt{\frac{d + \log(1/\delta)}{\sigma\,|\Omega_s^{d_k}|}} + \varepsilon_{\text{approx}}^s\right), \tag{25}$$

*where $C > 0$ is a constant depending on $B_\phi$, $B_r$, and the conditioning of the local Gram matrix, and $\varepsilon_{\text{approx}}^s$ is the local approximation error defined in Definition 3.1.*

*Proof.* Fix a query state $s$ and its ideal local transition set $\Omega_s^*$. By Definition 3.1, there exists a weight vector $w_s^*$ such that

$$Q_{\Omega_s^{d_k}}(s, a) = w_s^{*\top}\phi(s, a) + \varepsilon_s(s, a), \qquad |\varepsilon_s(s, a)| \leq \varepsilon_{\text{approx}}^s \tag{26}$$

for all $(s, a, r, s', a') \in \Omega_s^{d_k}$. By Assumption 3.3, the retrieved set $\Omega_s^{d_k}$ overlaps with the ideal set on at least $m = \sigma|\Omega_s^{d_k}|$ transitions. Denote this intersection by $\mathcal{D}_s^\sigma = \Omega_s^{d_k} \cap \Omega_s^*$. Thus, the estimation of $w_s^*$ from $\Omega_s^{d_k}$ is guaranteed to include at least $m$ valid local transitions. Let $X \in \mathbb{R}^{m\times d}$ be the feature matrix of $\mathcal{D}_s^\sigma$, with columns $\phi(\bar{s}, \bar{a})$, and let $y \in \mathbb{R}^m$ be the corresponding targets. Then

$$y = w_s^{*\top}X + \xi, \tag{27}$$

where $\xi$ collects the local approximation error, with $\|\xi\|_\infty \leq \varepsilon_{\text{approx}}^s$. The estimator from the retrieved set is

$$w_s(\Omega_s^{d_k}) = \arg\min_w \frac{1}{|\Omega_s^{d_k}|} \sum_{(s_i, a_i) \in \Omega_s^{d_k}} \left(y_i - w^\top \phi(s_i, a_i)\right)^2. \tag{28}$$

Define the population moments on $\Omega_s^*$ as

$$G = \mathbb{E}_{\Omega_s^*}[\phi^\top \phi], \quad b = \mathbb{E}_{\Omega_s^*}[\phi^\top y]. \tag{29}$$

Let $\hat{G}, \hat{b}$ be the corresponding empirical moments on $\Omega_s^{d_k}$. Since at least $m = \sigma|\Omega_s^*|$ samples in $\Omega_s^{d_k}$ come from the true local set, standard matrix concentration implies that, with probability at least $1 - \delta$,

$$\|\hat{G} - G\| \leq c_1 B_\phi^2 \sqrt{\frac{d + \log(1/\delta)}{\sigma |\Omega_s^{d_k}|}}, \tag{30}$$

$$\|\hat{b} - b\| \leq c_2 B_\phi B_r \sqrt{\frac{d + \log(1/\delta)}{\sigma |\Omega_s^{d_k}|}}, \tag{31}$$

for universal constants $c_1, c_2 > 0$. The optimal weight satisfies $w_s^{*\top} G = b$. The empirical solution satisfies $w_s(\Omega_s^{d_k})^\top \hat{G} = \hat{b}$ (up to residuals). Subtracting these systems gives

$$\|w_s(\Omega_s^{d_k}) - w_s^*\| \leq \|G^{-1}\| \cdot \left(\|\hat{b} - b\| + \|\hat{G} - G\|\|w_s^*\|\right) + \varepsilon_{\text{approx}}^s. \tag{32}$$

Since $G$ is well-conditioned, $\|G^{-1}\| \leq 1/\mu$ for some $\mu > 0$. Substituting the concentration results yields

$$\|w_s(\Omega_s^{d_k}) - w_s^*\| \leq C\sqrt{\frac{d + \log(1/\delta)}{\sigma |\Omega_s^{d_k}|}} + \varepsilon_{\text{approx}}^s, \tag{33}$$

where $C > 0$ depends on $B_\phi$, $B_r$, $\|w_s^*\|$, and $\mu$. This is exactly the desired bound in equation 25. $\quad\square$

**Theorem E.2** (Pointwise Q-function Error). *Suppose Assumption 3.1 and Assumption 3.3 hold. For any fixed $s \in \mathcal{S}$, with probability at least $1 - \delta$, the pointwise error of the estimated Q-function satisfies*

$$\left|\hat{Q}(s, a|\Omega_s^{d_k}) - Q_{\Omega_s^{d_k}}(s, a)\right| \leq \varepsilon_{\text{approx}}^s(1 + B_\phi) + CB_\phi \sqrt{\frac{d + \log(1/\delta)}{\sigma |\Omega_s^{d_k}|}} \quad \forall (s, a, r, s', a') \in \Omega_s^{d_k}, \tag{34}$$

*where $C > 0$ depends on $B_\phi$, $B_r$, and the conditioning of the local Gram matrix.*

*Proof.* Fix $s \in \mathcal{S}$ and $a \in \mathcal{A}$. By definition,

$$\hat{Q}(s, a|\Omega_s^{d_k}) = w_s(\Omega_s^{d_k})^\top \phi(s, a), \quad Q_{\Omega_s^{d_k}}(s, a) = w_s^{*\top} \phi(s, a) + \varepsilon_{\text{approx}}^s. \tag{35}$$

Thus,

$$\left|\hat{Q}(s, a|\Omega_s^{d_k}) - Q_{\Omega_s^{d_k}}(s, a)\right| = \left|w_s(\Omega_s^{d_k})^\top \phi(s, a) - w_s^{*\top} \phi(s, a) - \varepsilon_{\text{approx}}^s\right| \tag{36}$$

$$\leq \|w_s(\Omega_s^{d_k}) - w_s^*\| \cdot \|\phi(s, a)\| + \varepsilon_{\text{approx}}^s \tag{37}$$

$$\leq B_\phi \cdot \|w_s(\Omega_s^{d_k}) - w_s^*\| + \varepsilon_{\text{approx}}^s. \tag{38}$$

By Theorem E.1, with probability at least $1 - \delta$,

$$\|w_s(\Omega_s^{d_k}) - w_s^*\| \leq C\sqrt{\frac{d + \log(1/\delta)}{\sigma |\Omega_s^{d_k}|}} + \varepsilon_{\text{approx}}^s. \tag{39}$$

Substituting this inequality into the bound above yields

$$\left|\hat{Q}(s, a|\Omega_s^{d_k}) - Q_{\Omega_s^{d_k}}(s, a)\right| \leq CB_\phi \sqrt{\frac{d + \log(1/\delta)}{\sigma |\Omega_s^{d_k}|}} + \varepsilon_{\text{approx}}^s(1 + B_\phi), \tag{40}$$

which holds for all $(s, a, r, s', a') \in \Omega_s^{d_k}$. This proves equation 34. $\quad\square$

**Corollary E.3** (Expected Q-function Error). *Suppose Assumptions 3.1 and 3.3 hold. Let $\mu$ be a reference distribution over $(s,a) \in \mathcal{S} \times \mathcal{A}$, and let $\mu_{\mathcal{S}}$ be its marginal over states. Then, with probability at least $1 - \delta$, the expected Q-function approximation error restricted to the retrieved set satisfies*

$$\mathbb{E}_{(s,a)\sim\mu}\left[\left|\hat{Q}(s,a|\Omega_s^{d_k}) - Q_{\Omega_s^{d_k}}(s,a)\right| \mid (s,a) \in \Omega_s^{d_k}\right]$$
$$\leq \mathbb{E}_{s\sim\mu_{\mathcal{S}}}\left[\varepsilon_{approx}^s(1 + B_\phi) + CB_\phi\sqrt{\frac{d + \log(1/\delta)}{\sigma\,|\Omega_s^{d_k}|}}\right]. \tag{41}$$

*Proof.* From Theorem E.2, for any $(s,a,r,s',a') \in \Omega_s^{d_k}$, we have

$$\left|\hat{Q}(s,a|\Omega_s^{d_k}) - Q_{\Omega_s^{d_k}}(s,a)\right| \leq \varepsilon_{approx}^s(1 + B_\phi) + CB_\phi\sqrt{\frac{d+\log(1/\delta)}{\sigma|\Omega_s^{d_k}|}}. \tag{42}$$

Taking expectation over $(s,a) \sim \mu$, restricted to $(s,a) \in \Omega_s^{d_k}$, and noting that the right-hand side depends only on $s$, we obtain

$$\mathbb{E}_{(s,a)\sim\mu}\left[\left|\hat{Q}(s,a|\Omega_s^{d_k}) - Q_{\Omega_s^{d_k}}(s,a)\right| \mid (s,a) \in \Omega_s^{d_k}\right]$$
$$\leq \mathbb{E}_{s\sim\mu_{\mathcal{S}}}\left[\varepsilon_{approx}^s(1 + B_\phi) + CB_\phi\sqrt{\frac{d+\log(1/\delta)}{\sigma|\Omega_s^{d_k}|}}\right]. \tag{43}$$

This proves the result. □

### E.1 Proof of the Theorem on Policy Performance Gap

In this section, we provide the proof of Theorem 3.5.

**Lemma E.4** (Performance Difference Lemma). *Let $\pi$ be a policy, and let $d^\pi$ denote its discounted state distribution. Then the performance gap between $\pi$ and the optimal policy $\pi^*$ satisfies*

$$J(\pi^*) - J(\pi) = \frac{1}{1-\gamma}\,\mathbb{E}_{s\sim d^\pi, a\sim\pi}\left[Q^*(s,a^*) - Q^*(s,a)\right], \tag{44}$$

*where $a^* = \arg\max_a Q^*(s,a)$.*

*Proof.* From Equation (44), for any $s \in \mathcal{S}$,

$$Q^*(s,\pi^*(s)) - Q^*(s,\pi(s)) = \left(Q^*(s,\pi^*(s)) - \hat{Q}(s,\pi^*(s))\right) + \left(\hat{Q}(s,\pi^*(s)) - \hat{Q}(s,\pi(s))\right)$$
$$+ \left(\hat{Q}(s,\pi(s)) - Q^*(s,\pi(s))\right). \tag{45}$$

Since $\pi$ is greedy with respect to $\hat{Q}$, the middle term is non-positive. Thus,

$$Q^*(s,\pi^*(s)) - Q^*(s,\pi(s)) \leq |Q^*(s,\pi^*(s)) - \hat{Q}(s,\pi^*(s))| + |Q^*(s,\pi(s)) - \hat{Q}(s,\pi(s))|$$
$$\leq 2\delta(s), \tag{46}$$

where, by Theorem E.2,

$$\delta(s) = \varepsilon_{approx}^s(1 + B_\phi) + CB_\phi\sqrt{\frac{d+\log(1/\delta)}{\sigma|\Omega_s^{d_k}|}}. \tag{47}$$

Taking expectations in Equation (44) and applying Equation (46) yields

$$J(\pi^*) - J(\pi) \leq \frac{2}{1-\gamma}\,\mathbb{E}_{s\sim d^\pi}\left[\delta(s)\right], \tag{48}$$

which gives the desired bound in Equation (12). □

# F  ICQL VARIANTS FOR TD3+BC

In this section, we illustrate how to extend our method to TD3+BC (Fujimoto & Gu, 2021). TD3+BC introduces a simple behavior cloning regularization into value-based learning. This algorithm is easy to integrate with our framework, remains stable across diverse tasks, and serves as a strong baseline in the literature. Its simplicity and effectiveness make it an ideal testbed for evaluating the impact of localized Q-function estimation. Together, these properties provide sufficient coverage of common design choices in offline RL. Other algorithms can be extended in a similar manner, but we omit them here for clarity and focus.

Our proposed ICQL can be seamlessly integrated into existing offline RL algorithms by replacing the global Q-function with a local, context-dependent estimator defined in Definition 3.1. We demonstrate this idea by instantiating ICQL with TD3+BC; see Algorithm 1 for additional details.

**ICQL-TD3+BC.**  TD3+BC uses a standard Bellman backup for the critic and augments the actor objective with behavior cloning. We again use the locally estimated $\hat{Q}(s, a)$ in both components. The critic loss is:

$$\mathcal{L}_{\text{critic}}^{\text{TD3+BC}} = \mathbb{E}_{(s,a,r,s')\sim\mathcal{D}}\left[\left(\hat{Q}(s,a|\Omega_s^{d_k}) - y\right)^2\right], \tag{49}$$

where $y = r + \gamma\min_{i=1,2}\hat{Q}_{\text{target}}^{(i)}(s', \pi(s')|\Omega_s^{d_k})$. The actor is trained to maximize the estimated Q-value while remaining close to the dataset policy:

$$\mathcal{L}_{\text{actor}}^{\text{TD3+BC}} = -\mathbb{E}_{s\sim\mathcal{D}}\left[\hat{Q}(s,\pi(s)|\Omega_s^{d_k})\right] + \alpha\cdot\mathbb{E}_{(s,a)\sim\mathcal{D}}\left[\|\pi(s) - a\|^2\right]. \tag{50}$$

Experimental results are reported in Table 4.

Table 4: Evaluation for TD3+BC based ICQL variant on Mujoco and Adroit tasks. Average normalized scores are reported over 5 random seeds.

| Mujoco Tasks | TD3-BC | ICQL-TD3-BC(ours) | Gain(%) |
|---|---|---|---|
| Walker2d-Medium-Expert-v2 | 109.2 | **109.3** | 0.1% |
| Walker2d-Medium-v2 | **77.0** | 72.7 | -5.7% |
| Walker2d-Medium-Replay-v2 | 41.5 | **55.0** | 32.5% |
| Hopper-Medium-Expert-v2 | 78.2 | **87.2** | 11.5% |
| Hopper-Medium-v2 | 53.5 | **57.9** | 8.3% |
| Hopper-Medium-Replay-v2 | 59.4 | **65.8** | 10.9% |
| HalfCheetah-Medium-Expert-v2 | 62.8 | **63.7** | 1.5% |
| HalfCheetah-Medium-v2 | **43.1** | 42.7 | -0.8% |
| HalfCheetah-Medium-Replay-v2 | 41.8 | **45.9** | 9.8% |
| **Average** | 62.9 | **66.7** | 6.0% |
| **Adroit Tasks** | **TD3-BC** | **ICQL-TD3-BC(ours)** | **Gain(%)** |
| Pen-Human-v1 | 64.6 | **68.3** | 5.7% |
| Pen-Cloned-v1 | **76.8** | 74.7 | -2.8% |
| Hammer-Human-v1 | 1.5 | **1.6** | 7.9% |
| Hammer-Cloned-v1 | 1.8 | **7.3** | 300.6% |
| Door-Human-v1 | 0.2 | **2.0** | 1253.3% |
| Door-Cloned-v1 | **-0.1** | -0.1 | -60.0% |
| **Average** | 24.2 | **25.6** | 6.2% |

# G  IMPLEMENTATION DETAILS

In this section, we present the detailed network architectures of our in-context critic and actor. We also describe the hyperparameter settings used in this paper.

## G.1 IN-CONTEXT CRITIC NETWORK

The in-context critic consists of a feature extractor and a linear Transformer. The feature extractor is a 3-layer MLP with 256 hidden units. A Tanh activation is applied in the last layer, while ReLU is used in the other layers, followed by layer normalization. The output dimension of the feature extractor is 64. A dropout rate of 0.1 is applied during training. The linear Transformer is constructed as described in Equation (18), where trainable parameters appear only in $G$. The definition of $G$ is given in Equation (6), where $C_l$ denotes the trainable parameters in the $l$-th layer. The shape of $C_l$ is $64 \times 64$. We use gradient normalization to stabilize training by scaling gradients to have a maximum $L_2$ norm of 10. The number of linear Transformer layers is set to 20.

## G.2 POLICY NETWORK

For `ICQL-IQL`, the policy network is implemented as an MLP with 2 hidden layers and ReLU activations. The policy network also includes an additional learnable vector that represents the logarithmic standard deviation of the action distribution. A dropout rate of 0.1 is applied during training.

For `ICQL-TD3+BC`, the policy network is implemented as a 3-layer MLP with ReLU activations.

## G.3 HYPER-PARAMETER SETTINGS

For `ICQL-IQL`, we follow the original IQL paper and use different values of the expectile parameter $\tau$ and the temperature parameter $\beta$ for different offline datasets. We search over $\{0.5, 0.7, 0.9\}$ for the expectile parameter and $\{1, 2, 3\}$ for the temperature parameter. The detailed settings are listed in Table 5.

Table 5: Expectile and temperature settings for `ICQL` experiments.

| Tasks | Expectile | Temperature | Tasks | Expectile | Temperature |
|---|---|---|---|---|---|
| Walker2d-Medium-Expert-v2 | 0.7 | 1 | Pen-Human-v1 | 0.7 | 2 |
| Walker2d-Medium-v2 | 0.7 | 1 | Pen-Cloned-v1 | 0.9 | 2 |
| Walker2d-Medium-Replay-v2 | 0.7 | 1 | Hammer-Human-v1 | 0.5 | 1 |
| Hopper-Medium-Expert-v2 | 0.7 | 1 | Hammer-Cloned-v1 | 0.9 | 2 |
| Hopper-Medium-v2 | 0.5 | 1 | Door-Human-v1 | 0.5 | 1 |
| Hopper-Medium-Replay-v2 | 0.7 | 2 | Door-Cloned-v1 | 0.7 | 2 |
| HalfCheetah-Medium-Expert-v2 | 0.5 | 2 | Kitchen-Complete-v0 | 0.9 | 1 |
| HalfCheetah-Medium-v2 | 0.5 | 1 | Kitchen-Mixed-v0 | 0.5 | 1 |
| HalfCheetah-Medium-Replay-v2 | 0.7 | 1 | Kitchen-Partial-v0 | 0.9 | 2 |

For `ICQL-TD3+BC`, we follow the settings in the original paper and use the same hyperparameter value $\alpha = 2.5$ for all datasets.

Other common hyperparameters are listed in Table 6.

Table 6: Common hyperparameters for the main `ICQL` experiments.

| Hyperparameter | Value |
|---|---|
| Hidden dimension | 256 |
| Batch size | 256 |
| Training steps | 1,000,000 |
| Evaluation episodes | 10 |
| Discount factor | 0.99 |
| Policy learning rate | 3.0e-4 |
| Critic learning rate | 3.0e-4 |
| Context length | 20 |

# H ADDITIONAL EXPERIMENT RESULTS

## H.1 EXTENDED BASELINES

In this section, we extend our comparisons to additional methods, including RA-DT Schmied et al. (2024), ReBRAC Tarasov et al. (2023), DMG Mao et al. (2024), FQL Park et al. (2025), and QC Li et al. (2025), following their official implementations. ICQL demonstrates competitive or superior performance on most tasks. The results are shown in Table 7.

Table 7: Performance comparison across Mujoco, Adroit, and Kitchen tasks. Average and standard deviation of scores are reported over 5 random seeds.

| Task | BC | TD3BC | CQL | IQL | DT | RADT | ReBRAC | DMG | FQL | QC | ICQL |
|------|----|-------|-----|-----|----|----|--------|-----|-----|-----|------|
| Walker2d-ME | 107.5 | 109.2 | 98.7 | 109.8 | 70.7 | 107.8 | 109.2 | 109.5 | 101.0 | 102.8 | **113.3** |
| Walker2d-M | 75.3 | 77.0 | 79.2 | 71.5 | 70.2 | 68.9 | 82.8 | **85.0** | 72.4 | 34.1 | 80.3 |
| Walker2d-MR | 26.0 | 41.5 | 77.2 | 61.0 | 54.8 | 67.2 | 39.4 | 81.9 | 60.9 | 46.6 | **81.9** |
| Hopper-ME | 52.5 | 78.2 | 105.4 | 98.5 | 57.5 | 109.4 | 98.7 | 109.8 | 60.1 | 44.0 | **111.0** |
| Hopper-M | 52.9 | 53.5 | 58.0 | 63.3 | 57.1 | 62.4 | 60.6 | **92.3** | 55.6 | 64.6 | 62.6 |
| Hopper-MR | 18.1 | 59.4 | 95.0 | 82.4 | 65.8 | 81.6 | 87.4 | **100.1** | 55.0 | 18.6 | 96.4 |
| HalfCheetah-ME | 55.2 | 62.8 | 62.4 | 83.4 | 70.8 | 90.9 | 84.6 | 93.6 | 92.9 | **94.2** | 89.1 |
| HalfCheetah-M | 42.6 | 43.1 | 44.4 | 42.5 | 42.8 | 42.0 | 44.6 | 47.9 | 43.9 | **48.2** | 45.9 |
| HalfCheetah-MR | 36.6 | 41.8 | **45.5** | 38.9 | 39.5 | 38.9 | 36.9 | 44.6 | 40.0 | 40.5 | 44.7 |
| Pen-Human | 63.9 | 64.6 | 37.5 | 89.5 | 79.5 | 17.8 | **91.5** | 66.2 | 61.2 | 55.7 | 85.6 |
| Pen-Cloned | 37.0 | 76.8 | 39.2 | 84.9 | 74.0 | 32.4 | 68.9 | 67.5 | 23.5 | 54.8 | **89.4** |
| Hammer-Human | 1.2 | 1.5 | 4.4 | 7.2 | 1.7 | 0.7 | 1.1 | **18.4** | 1.1 | 1.2 | 3.7 |
| Hammer-Cloned | 0.6 | 1.8 | 2.1 | 0.5 | 3.7 | 1.3 | 0.2 | **13.4** | 1.7 | 2.2 | 4.5 |
| Door-Human | 2.0 | 0.2 | 9.9 | 9.8 | 5.5 | 13.2 | -0.1 | 0.1 | 0.2 | 0.7 | **17.1** |
| Door-Cloned | 0.0 | -0.1 | 0.1 | 7.6 | 3.2 | 2.4 | 9.0 | 3.7 | 0.1 | 4.4 | **11.7** |
| Kitchen-Complete | 65.0 | 57.5 | 43.8 | 59.2 | 52.5 | 32.5 | 60.0 | 22.5 | 16.3 | 27.5 | **79.3** |
| Kitchen-Mixed | 51.5 | 53.5 | 51.0 | 53.3 | **60.0** | 54.1 | 47.5 | 30.0 | 45.0 | 60.0 | 59.5 |
| Kitchen-Partial | 38.0 | 46.7 | 49.8 | 45.8 | 55.0 | 53.8 | 62.5 | 37.5 | 15.8 | 52.5 | **61.5** |
| **Overall Average** | 47.3 | 47.7 | 58.5 | 63.2 | 56.2 | 57.2 | 60.0 | 62.0 | 50.5 | 47.7 | **69.7** |

## H.2 NUMERICAL RESULTS FOR ABLATION STUDIES ON THE NUMBER OF LAYERS AND CONTEXT LENGTHS

In this section, we provide numerical results corresponding to Section 4.3.1 and Section 4.3.2.

Table 8: Normalized scores for Gym tasks with different lengths of contexts and different number of layers in `ICQL-IQL`.

| | Context Length | | | | Number of Layers | | | |
|------|-----|-----|-----|-----|-----|-----|-----|-----|
| **Gym Tasks** | **10** | **20** | **30** | **40** | **4** | **8** | **16** | **20** |
| Walker2d-Medium-Expert | 111.1 | **113.3** | 111.7 | 110.2 | 102.3 | 103.3 | 104.1 | **113.3** |
| Walker2d-Medium | 79.6 | 80.3 | 70.9 | **80.7** | 78.0 | 78.4 | 74.9 | **80.3** |
| Walker2d-Medium-Replay | 77.5 | **81.9** | 69.4 | 74.4 | 76.3 | 77.0 | 75.8 | **81.9** |
| Hopper-Medium-Expert | 103.7 | **111.0** | 106.0 | 103.4 | 104.8 | **111.8** | 107.0 | 111.0 |
| Hopper-Medium | **73.8** | 62.6 | 60.2 | 59.4 | 65.7 | **67.6** | 67.3 | 62.6 |
| Hopper-Medium-Replay | 89.9 | **96.4** | 81.2 | 83.9 | **100.5** | 97.8 | 91.8 | 96.4 |
| HalfCheetah-Medium-Expert | **89.2** | 89.1 | 88.8 | 83.5 | 71.3 | 63.3 | 74.8 | **89.1** |
| HalfCheetah-Medium | 45.9 | 45.9 | **46.3** | 45.8 | 45.1 | 44.8 | 45.0 | **45.9** |
| HalfCheetah-Medium-Replay | 43.7 | **44.7** | 44.3 | 44.2 | 43.5 | 43.6 | 43.8 | **44.7** |
| **Average** | 79.4 | **80.6** | 75.4 | 76.2 | 76.4 | 76.4 | 76.0 | **80.6** |

## H.3 Computation Overhead Analysis

### H.3.1 Comparison of Training Time, Inference Time, GFLOPs, and Memory Consumption

In this section, we compare training time, inference time, GFLOPs, and memory consumption across all baseline methods. The analysis is conducted on the Walker2d-Medium-Expert dataset, and the results are summarized in Table 9. This analysis shows that, although ICQL incurs moderate additional computational cost relative to most advanced baselines, it remains more efficient than sequential models such as DT and RA-DT while achieving substantially stronger performance.

Table 9: Computation cost comparison across offline RL algorithms, including per-step training/inference time, FLOPs, and peak memory consumption.

| Algorithm | Train Time (ms) | Infer Time (ms) | Training GFLOPs | Peak Memory (MB) |
|---|---|---|---|---|
| TD3BC | 7.23 | 0.26 | 0.17 | 30 |
| IQL | 10.52 | 0.61 | 0.22 | 26 |
| CQL | 47.57 | 0.61 | 2.64 | 79 |
| DT | 68.42 | 2.89 | 151.40 | 1383 |
| RA-DT | 121.02 | 3.13 | 1103.79 | 1424 |
| ReBRAC | 13.91 | 0.26 | 0.18 | 38 |
| DMG | 32.33 | 0.42 | 0.55 | 27 |
| FQL | 19.63 | 0.37 | 4.53 | 126 |
| QC | 21.60 | 0.25 | 4.65 | 244 |
| ICQL | 70.73 | 0.51 | 1.03 | 375 |

### H.3.2 Analysis of GFLOPs and Memory Consumption Scaling of ICQL

We further report training GFLOPs and memory consumption for context lengths in $\{10, 20, 30, 40\}$ and for different numbers of linear Transformer layers in Table 10 and Table 11. The required training time scales with both context length and the number of layers. Using a context length of 20 and 20 linear Transformer layers remains comparatively efficient while providing competitive performance.

Table 10: Training FLOPs (in GFLOPs) for different numbers of layers and context lengths $K$.

| # Layers | K=10 | K=20 | K=30 | K=40 |
|---|---|---|---|---|
| 10 | 0.25 | 0.51 | 0.81 | 1.14 |
| 20 | 0.50 | 1.03 | 1.62 | 2.28 |
| 30 | 0.75 | 1.54 | 2.43 | 3.42 |
| 40 | 1.00 | 2.06 | 3.24 | 4.56 |

Table 11: Peak memory consumption (in MB) for different numbers of layers and context lengths $K$.

| # Layers | K=10 | K=20 | K=30 | K=40 |
|---|---|---|---|---|
| 10 | 171.28 | 306.56 | 445.57 | 590.39 |
| 20 | 209.71 | 375.38 | 549.51 | 738.00 |
| 30 | 248.39 | 443.29 | 655.26 | 879.30 |
| 40 | 288.94 | 511.58 | 758.94 | 1023.75 |

### H.3.3 Detailed Comparison of Retrieval and Training Time of ICQL across All Datasets

To mitigate repeated computation, we pre-compute all retrieval indices once before training for three reasons: (1) the offline dataset is fixed; (2) the retrieval rule is deterministic; and (3) pre-computation does not affect the learning dynamics or outcomes. This converts the per-step retrieval cost into

an amortized constant-time lookup during training. ICQL follows a standard actor-critic training paradigm in which the critic uses retrieved context to estimate local Q-values and the policy learns from these Q-values. At evaluation time, only the policy is used, which is consistent with standard actor-critic practice.

We report the real-time retrieval cost, the lookup time with cached indices, and the training and inference speed for all datasets. The reported results are averaged over all datasets used in our experiments. As shown in Table 12, cached retrieval adds only approximately 0.03 ms per step, which is negligible relative to the overall training time. A detailed breakdown of retrieval time and training and inference time is provided in Table 13 and Table 14.

Table 12: Average ICQL runtime of retrieval, training with different context lengths, and inference, across all datasets.

|  | Time (ms) |
| --- | --- |
| **Retrieval with Cached Index** | 0.03 |
| **Train with K=10** | 46.94 |
| **Train with K=20** | 72.15 |
| **Train with K=30** | 113.86 |
| **Train with K=40** | 171.95 |
| **Inference** | 0.54 |

Table 13: Detailed retrieval time (ms) analysis across tasks and context lengths. Cached index retrieval eliminates repeated nearest-neighbor searches and greatly reduces overhead.

| Task | Dataset Size | K=10 | K=20 | K=30 | K=40 | Cached |
| --- | --- | --- | --- | --- | --- | --- |
| Walker2d-Medium-Expert | 1998318 | 6.38 | 6.52 | 6.90 | 7.70 | 0.04 |
| Walker2d-Medium | 999322 | 3.98 | 3.96 | 4.37 | 5.16 | 0.03 |
| Walker2d-Medium-Replay | 301698 | 1.92 | 2.18 | 2.64 | 3.90 | 0.03 |
| Hopper-Medium-Expert | 1998966 | 6.04 | 6.11 | 6.39 | 7.31 | 0.03 |
| Hopper-Medium | 999998 | 3.85 | 3.71 | 4.05 | 4.77 | 0.03 |
| Hopper-Medium-Replay | 401598 | 2.10 | 2.16 | 2.56 | 3.11 | 0.03 |
| HalfCheetah-Medium-Expert | 1998000 | 6.27 | 6.41 | 6.75 | 7.37 | 0.03 |
| HalfCheetah-Medium | 999000 | 3.96 | 3.90 | 4.24 | 5.17 | 0.04 |
| HalfCheetah-Medium-Replay | 201798 | 1.58 | 1.61 | 1.81 | 2.53 | 0.03 |
| Pen-Human | 4975 | 0.89 | 0.81 | 0.99 | 1.15 | 0.03 |
| Pen-Cloned | 496264 | 2.91 | 3.05 | 3.52 | 6.67 | 0.03 |
| Hammer-Human | 11285 | 0.88 | 0.89 | 1.05 | 1.17 | 0.03 |
| Hammer-Cloned | 996394 | 4.56 | 4.54 | 4.93 | 5.82 | 0.03 |
| Door-Human | 6704 | 0.88 | 0.88 | 1.07 | 1.17 | 0.03 |
| Door-Cloned | 995642 | 4.39 | 4.53 | 4.92 | 5.94 | 0.03 |
| Kitchen-Complete | 3679 | 0.89 | 0.82 | 0.95 | 1.12 | 0.03 |
| Kitchen-Partial | 136937 | 1.36 | 1.41 | 1.60 | 1.91 | 0.03 |
| Kitchen-Mixed | 136937 | 1.49 | 1.38 | 1.63 | 2.11 | 0.03 |

### H.4  FAILURE ANALYSIS ON THE HAMMER DATASET

In this section, we provide a failure analysis on the Hammer-Human dataset. We find that Hammer-Human exhibits two properties that make it particularly challenging for ICQL.

First, the dataset size is small and the coverage is sparse. Hammer-Human contains only 24 trajectories (~11k transitions), which is substantially fewer than Hammer-Cloned (~996k transitions). This leads to larger distances between the query state and its retrieved neighbors, which violates locality assumptions, and to poorer state-space coverage, which makes retrieval more likely to include semantically irrelevant transitions.

Table 14: Training and inference time (ms) for different context lengths across tasks. Training time grows approximately linearly with the context length, while inference time remains nearly constant.

| Task | K=10 | K=20 | K=30 | K=40 | Inference |
|---|---|---|---|---|---|
| Walker2d-Medium-Expert | 48.90 | 70.73 | 111.75 | 170.71 | 0.51 |
| Walker2d-Medium | 46.63 | 71.77 | 113.82 | 170.85 | 0.50 |
| Walker2d-Medium-Replay | 48.68 | 74.75 | 115.43 | 171.97 | 0.52 |
| Hopper-Medium-Expert | 48.31 | 70.71 | 114.56 | 173.52 | 0.51 |
| Hopper-Medium | 46.39 | 71.60 | 113.35 | 171.63 | 0.57 |
| Hopper-Medium-Replay | 46.08 | 72.32 | 112.89 | 170.58 | 0.56 |
| HalfCheetah-Medium-Expert | 48.33 | 73.27 | 115.90 | 171.46 | 0.58 |
| HalfCheetah-Medium | 47.69 | 74.45 | 113.85 | 171.75 | 0.51 |
| HalfCheetah-Medium-Replay | 47.30 | 71.81 | 114.32 | 172.86 | 0.57 |
| Pen-Human | 45.65 | 72.41 | 114.23 | 171.73 | 0.56 |
| Pen-Cloned | 44.50 | 69.59 | 112.02 | 170.31 | 0.51 |
| Hammer-Human | 46.88 | 73.78 | 113.93 | 171.66 | 0.52 |
| Hammer-Cloned | 47.31 | 72.55 | 114.13 | 171.94 | 0.57 |
| Door-Human | 46.16 | 71.34 | 113.11 | 171.44 | 0.57 |
| Door-Cloned | 45.37 | 71.20 | 112.11 | 170.61 | 0.58 |
| Kitchen-Complete | 47.45 | 73.65 | 116.36 | 175.98 | 0.54 |
| Kitchen-Partial | 48.11 | 72.35 | 116.50 | 175.42 | 0.52 |
| Kitchen-Mixed | 45.25 | 70.47 | 111.17 | 170.59 | 0.52 |

Second, the transitions are of low quality and the rewards are noisy. Most Hammer-Human trajectories have very low returns. As a result, for each query state, the retrieved neighbors tend to provide weak reward signals, which makes it more difficult to fit an effective local Q-function.

We provide comparisons of dataset statistics in Table 15 and comparisons of the distributions of the mean distance between query states and retrieved states in Figure 6. Both support these observations.

Table 15: Dataset statistics for Hammer-Human and Hammer-Cloned.

| Dataset | Hammer-Human | Hammer-Cloned |
|---|---|---|
| **Number of trajectories** | 24 | 3605 |
| **Number of transitions** | 11285 | 996394 |
| **Mean Trajectory Length (Min–Max)** | 455.2 (347–623) | 276.4 (199–623) |
| **Mean Trajectory Return (Min–Max)** | 2817.5 (-109–16022) | 779.8 (-407–16022) |

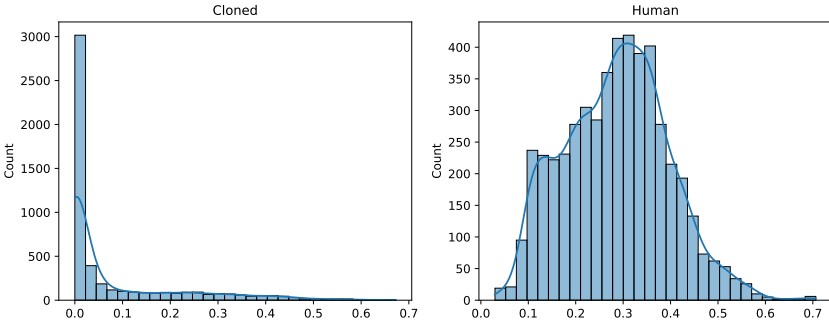

Figure 6: Distribution of mean distance between query states and retrieved states on Hammer dataset.

Although Hammer-Cloned also contains mostly low-return behavior, its much larger dataset size provides substantially denser coverage. As a result, ICQL can retrieve states that are much closer to the query state, which enables more reliable local linear approximation and yields slightly higher scores.

Moreover, for both Hammer-Human and Hammer-Cloned, the proportion of high-reward transitions is extremely low, which makes it inherently difficult to retrieve a local neighborhood that provides strong positive supervision. As a result, even if the Q-network successfully fits a local linear approximation, it rarely observes transitions that reliably correspond to high-return behavior. Consequently, the learned Q-values cannot meaningfully distinguish truly rewarding actions, which leads to uniformly low evaluation scores on both datasets.

### H.5 ANALYSIS OF THE RELATIONSHIP BETWEEN THEORETICAL $d$ AND $k$

In the theory, a local set $\Omega^d_{s_{query}}$ is defined as the set of all transitions whose states fall within a radius-$d$ neighborhood around $s$. This radius determines the intrinsic locality scale at which the Q-function is assumed to be approximately linear. However, in practice, the radius $d$ is not directly tunable: it depends on the underlying density and geometry of the dataset and is unknown to the algorithm.

Instead, ICQL controls locality through the retrieval size $k$. Retrieving the top-$k$ nearest neighbors is equivalent to selecting a data-adaptive radius, where $d_k = \max_{(s_i,\cdot)\in\text{top-}k} ||s_i - s||_2^2$ and $\bar{d}_k = \max_{(s'_i,\cdot)\in\text{top-}k} ||s'_i - s_i||_2^2$, so that the practical neighborhood is exactly the theoretical local set with radius $(d_k, \bar{d}_k)$. The distribution of the mean distance between query states and retrieved states for different values of $k$ is visualized in Figure 7.

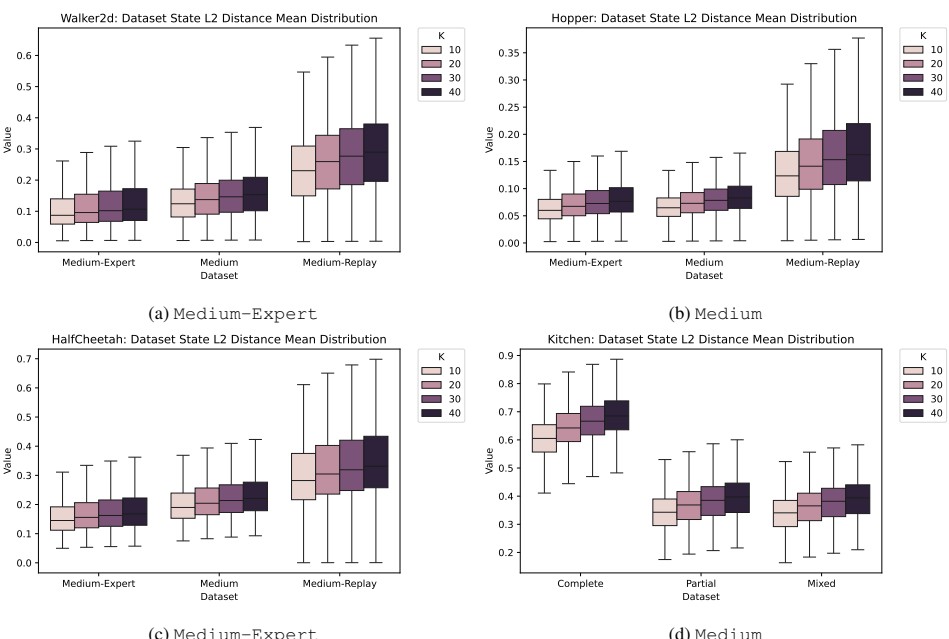

(a) Medium-Expert

(b) Medium

(c) Medium-Expert

(d) Medium

Figure 7: Distribution of mean distance between query states and retrieved states of different $k$.

Thus, $k$ determines the effective radius implicitly and monotonically: a larger $k$ expands the radius $(d_k, \bar{d}_k)$ and increases the size and heterogeneity of $\Omega^{d_k}_s$, whereas a smaller $k$ leads to tighter neighborhoods with more consistent local value structure.

### H.6 ADDITIONAL VISUALIZATION OF LEARNED Q-VALUE COMPARISON

We extend the visualization analysis of learned Q-values for ICQL and IQL by comparing them with Q-values learned by the online RL method SAC on the Walker2d-Medium-Expert, Walker2d-

Medium, and Walker2d-Medium-Replay datasets. We scale all Q estimates to the range [0,1] before visualization. We also include additional scatter plots that compare the Q-values estimated by each method against the SAC oracle. The visualizations are shown in Figure 8 and Figure 9. These plots clearly show that the correlation between ICQL and SAC is stronger than that between IQL and SAC, which indicates that ICQL produces more accurate value estimates than IQL.

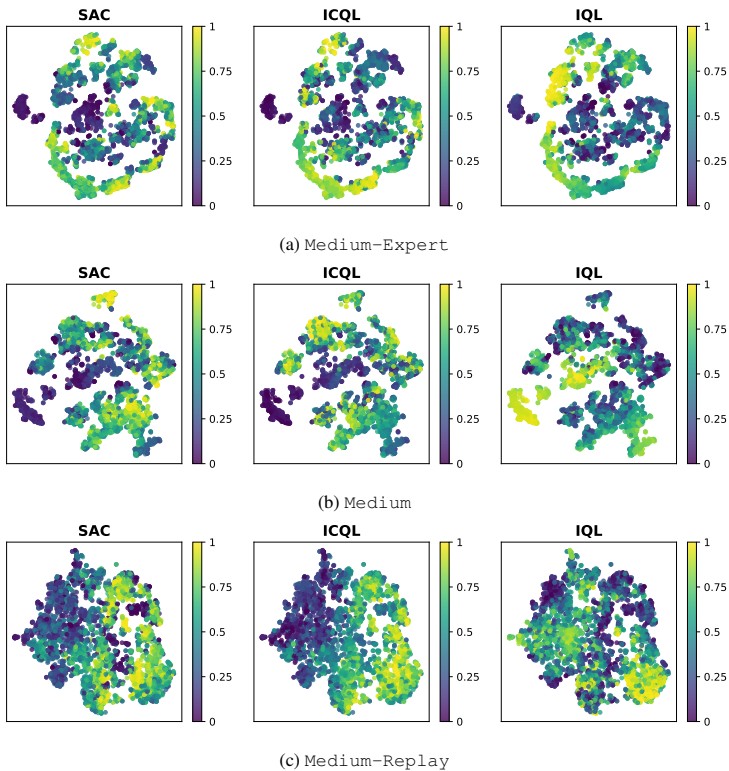

(a) `Medium-Expert`

(b) `Medium`

(c) `Medium-Replay`

Figure 8: Q-value of Walker2d-Medium-Expert, Walker2d-Medium, and Walker2d-Medium-Replay dataset on t-SNE mapped state distribution.

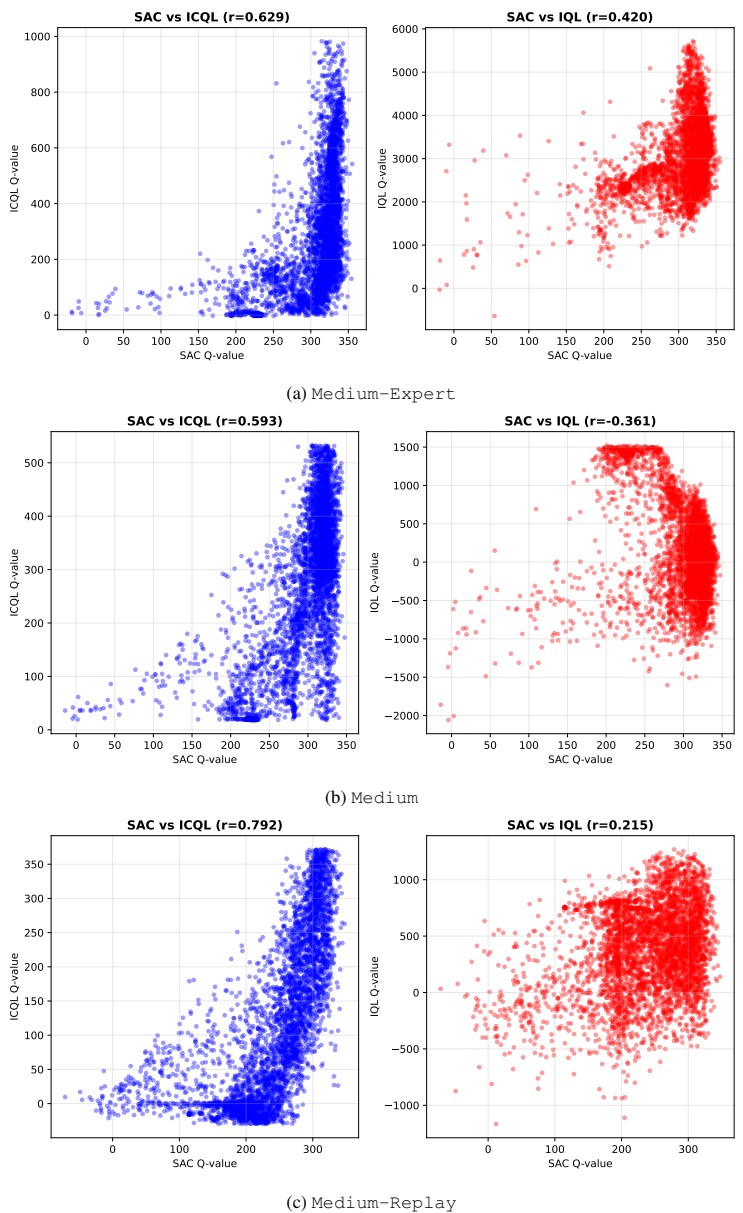

Figure 9: Q-value correlation of Walker2d-Medium-Expert, Walker2d-Medium, and Walker2d-Medium-Replay dataset. Red: Correlation between Q-values learned by IQL and SAC. Blue: Correlation between Q-values learned by ICQL and SAC.

## H.7 ANALYSIS OF IN-CONTEXT CRITICS

In this section, we conduct additional analysis of the functionality of our in-context Q estimator. By construction, the forward pass of our in-context Q estimator is equivalent to step-wise optimization of the TD error. We analyze the outputs and parameter distributions of each intermediate layer to validate its effectiveness. We randomly select 10 different states and their corresponding actions from the offline dataset of Walker2d-Medium-Expert-v2, retrieve 20 relevant transitions using cosine state similarity, and estimate the Q-values for these state-action pairs. We store the outputs of all intermediate layers, and the visualization results are shown in Figure 10. From Figure 10, we observe that the Q estimates exhibit a converging trend as the layers become deeper, which validates the iterative refinement process.

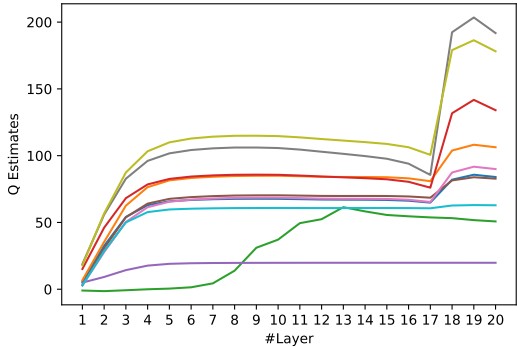

Figure 10: Q-estimates at each intermediate layer.

