# OpenReview forum: "In-Context Compositional Q-Learning for Offline  Reinforcement Learning"
_ICLR.cc/2026/Conference — ICLR 2026 Poster_

### Official Review · Reviewer_MZEj · 2025-10-23

**Soundness:** 3
**Presentation:** 3
**Contribution:** 3
**Rating:** 4
**Confidence:** 3

**Summary:**

This paper introduces ICQL , a novel offline RL framework that formulates Q-learning as a contextual reasoning problem. The core idea is to use a linear Transformer to learn local Q-functions from retrieved similar transition data, rather than fitting a single global Q-function. The authors argue that many tasks exhibit compositional structures and local value patterns, and they provide theoretical guarantees on bounded approximation error and near-optimal policy extraction. Experiments on the D4RL benchmark demonstrate performance improvements of 8.6% on Gym, 6.3% on Adroit, and 16.4% on Kitchen tasks.

**Strengths:**

Viewing value learning in offline RL as a In-contextual learning problem is highly innovative and well-motivated by the observation that the state space exhibits local compositional structure. The paper provides a formal analysis of bounded approximation error (Theorem 3.5) and connects retrieval quality to performance through a coverage assumption. Moreover, it successfully integrates with existing methods such as IQL and TD3+BC, demonstrating broad applicability of the proposed framework.

**Weaknesses:**

1.The local linear Q-function approximation (Definition 3.1) is a standard practice in the literature. The paper’s main theoretical contribution lies in connecting this formulation to in-context learning, but the underlying assumptions—particularly Assumption 3.3 on set coverage—are quite strong and may not hold in practical settings.

2.Retrieving k-nearest neighbors for each query and processing them with a multi-layer Transformer introduces significant computational overhead compared to standard methods. However, the paper does not provide any runtime analysis or comparison to quantify this cost.

3.Weak justification for using linear Transformers: Why specifically choose linear attention? Although the paper claims to reveal the “underexplored potential of linear attention,” it provides no comparison with standard self-attention, leaving the rationale insufficiently supported.

4.Limitations of the experimental evaluation: The experiments are conducted only on the D4RL benchmark, which is not sufficiently comprehensive. More importantly, the paper does not compare against recent state-of-the-art offline reinforcement learning methods, limiting the strength of its empirical validation.

**Questions:**

1.Has the paper conducted any comparison between linear Transformers and standard self-attention? Which specific properties of linear attention are deemed essential or indispensable for the proposed method’s effectiveness?

2.Assumption 3.3 requires knowledge of the coverage parameter σ, but it remains unclear how this parameter is determined in practice. How is σ set or estimated empirically, and how well does the theoretical analysis based on this assumption correlate with actual performance?

3.Why does ICQL perform catastrophically on the Hammer-Human task (−49.4%)? Is this performance drop solely due to dataset characteristics, or does it reveal a fundamental limitation of the proposed method? Clarifying this distinction would help assess the robustness and generality of ICQL.

4.The paper states that, "After training, the extracted policy can be evaluated on its own without extra retrieval process or
contextual inference." However, since the policy is trained using context-dependent Q-values, how does it function without access to retrieval during evaluation? Does this setup introduce a train–test mismatch or affect the consistency between training and inference behaviors?

5.Why does the paper not include comparisons with more recent approaches (e.g., [1, 2, 3, 4]) or other modern offline RL algorithms? Such comparisons are essential to demonstrate the competitiveness and relevance of the proposed method in the current research landscape.

[1]Tarasov, D., Kurenkov, V., Nikulin, A., & Kolesnikov, S. Revisiting the minimalist approach to offline reinforcement learning. NIPS， 2023.

[2]Mao Y, Wang Q, Qu Y, et al. Doubly mild generalization for offline reinforcement learning. NIPS, 2024.

[3]Park S, Li Q, Levine S. Flow q-learning. ICML, 2025.

[4]Li Q, Zhou Z, Levine S. Reinforcement learning with action chunking. arXiv preprint arXiv:2507.07969, 2025.

---

> ### Author Response · Authors · 2025-11-25
>
> We sincerely thank the reviewer for the thoughtful and encouraging evaluation of our work. We greatly appreciate your recognition of the paper’s key strengths. Below, we provide detailed responses to your questions and concerns.
>
> **W1. Relation between assumption 3.3 on set coverage and practical settings.**
>
> We sincerely thank the reviewer for highlighting this important point. Our intention is not to assume that such idealized coverage is readily satisfied in practice, but rather to use the assumption as a theoretical lens to illuminate how retrieval quality affects local value estimation within the in-context learning framework.
>
> Our empirical results consistently show that insufficient coverage (e.g., small context length or low-quality retrieval) correlates with degraded performance—precisely matching the theoretical implications of Assumption 3.3.
>
> **W2. Computational overhead of retrieval and linear transformer**
>
> We appreciate the reviewer’s constructive suggestion. As the offline dataset is fixed and the retrieval process is deterministic, we pre-compute the nearest-neighbor index before training, eliminating repeated distance computations during learning.
>
> The table below reports the average real-time retrieval time, the lookup time with cached indices, and the training speed. As shown, **cached retrieval adds only ~0.03 ms per step**, which is negligible relative to the overall training time.
>
> |  | **Time (ms)** |
> | --- | --- |
> |**Retrieval with Cached Index**| 0.03 |
> |**Train with K=10**|46.94|
> |**Train with K=20**|72.15|
> |**Train with K=30**|113.86|
> |**Train with K=40**|171.95|
> |**Inference**|0.54|
>
> We have also added a comprehensive compute analysis across all baselines on Walke2d-Medium-Expert dataset, as summarized below:
>
> | Algorithm | Avg Time per Training Step (ms) | Avg Time per Inference Step (ms) | Training FLOPs (GFLOPS) | Peak Memory (MB) |
> |-|-|-|-|-|
> |TD3BC|7.23|0.26|0.17|30|
> |IQL|10.52|0.61|0.22|26|
> |CQL|47.57|0.61|2.64|79|
> |DT|68.42|2.89|151.40|1383|
> |RA-DT|121.02|3.13|1103.79|1424|
> |ReBRAC|13.91|0.26|0.18|38|
> |DMG|32.33|0.42|0.55|27|
> |FQL|19.63|0.37|4.53|126|
> |QC|21.60|0.25|4.65|244|
> |ICQL|70.73|0.51|1.03|375|
>
> We further report training time consumption for varying context lengths in $\{10,20,30,40\}$:
>
> |**Context length**|**Avg Time per Training Step (ms)**|
> |-|-|
> |10|48.90|
> |20|70.73|
> |30|111.75|
> |40|170.71|
>
> These results show that while ICQL incurs moderate additional compute cost relative to most advanced baselines, and it remains more efficient than sequential models (DT/RADT) while achieving substantially stronger performance. The training time needed scales with context length, and using a context length of 20 remains comparable efficient while providing competitive performance. All the results have been added to Appendix I.4 (Page 26-28) of the revised manuscript.
>
> **W3&Q1. Justification for linear attention vs. standard attention**
>
> Thank you for raising this important question. Our choice of linear attention is motivated by both theoretical interpretability and validated by empirical performance.
>
> 1. **Theoretical interpretability**: Linear attention admits a closed-form interpretation as in-context TD learning, which is thoroughly proved by the literature[1]. This property allows us to view the linear attention head as implicitly implementing a local TD learner, aligning precisely with our theoretical formulation. Standard self-attention lacks this interpretability, making it less suitable for establishing the theoretical connection emphasized in our work.
> 2. **Empirical performance**: We have compared the performance between linear attention and standard self-attention with identical context size. Results are shown in the table below: linear attention substantially outperforms standard attention, indicating that the specific design of linear attention is essential.
> |**Task**|**Linear Attention**|**Standard Attention**|
> |-|-|-|
> |**Hopper-Medium**|62.6|61.9|
> |**Hopper-Medium-Expert**|113.3|70.3|
> |**Hopper-Medium-Replay**|96.4|42.1|
> |**HalfCheetah-Medium**|45.9|42.0|
> |**HalfCheetah-Medium-Expert**|89.1|72.5|
> |**HalfCheetah-Medium-Replay**|44.7|36.1|
> |**Walker2d-Medium**|80.3|77.4|
> |**Walker2d-Medium-Expert**|113.3|108.8|
> |**Walker2d-Medium-Replay**|81.9|42.9|
> |**Pen-Human**|85.6|72.7|
> |**Pen-clone**|89.4|83.8|
> |**Hammer-Human**|3.7|4.2|
> |**Hammer-clone**|4.5|1.8|
> |**Door-Human**|17.1|8.9|
> |**Door-Cloned**|11.7|3.4|
> |**Kitchen-Complete**|79.3|78.3|
> |**Kitchen-Mixed**|59.5|55.8|
> |**Kitchen-Partial**|61.5|55.8|
>
> Thus, linear attention is not only theoretically aligned with our framework but also provides the most stable and consistent empirical performance. We have added these results to the revised Appendix I.3 (Page 26).
>
> [1] Jiuqi Wang, Ethan Blaser, Hadi Daneshmand, and Shangtong Zhang. Transformers can learn temporal difference methods for in-context reinforcement learning. In The Thirteenth International Conference on Learning Representations, 2025.

---

> ### Author Response · Authors · 2025-11-25
>
> **W4&Q5. Experimental scope**
>
> We thank the reviewer for this valuable suggestion. D4RL benchmark is the standardized, widely adopted benchmark for offline RL research, so we mainly consider this benchmark.
>
> We fully agree that including more recent baselines strengthens our empirical evaluation. Accordingly, we have incorporated comparisons with the reviewer-suggested methods (ReBRAC, DMG, FQL, QC), following their official implementations. ICQL demonstrates competitive or superior performance across most tasks:
>
> | **Task** | **ReBRAC** | **DMG** | **FQL** | **QC** | **ICQL** |
> | --- | --- | --- | --- | --- | --- |
> | **Walker2d-Medium-Expert** | 109.2 | 109.5 | 101.0 | 102.8 | 113.3 |
> | **Walker2d-Medium** | 82.8 | 85.0 | 72.4 | 34.1 | 80.3 |
> | **Walker2d-Medium-Replay** | 39.4 | 81.9 | 60.9 | 46.6 | 81.9 |
> | **Hopper-Medium-Expert** | 98.7 | 109.8 | 60.1 | 44.0 | 113.3 |
> | **Hopper-Medium** | 60.6 | 92.3 | 55.6 | 64.6 | 62.6 |
> | **Hopper-Medium-Replay** | 87.4 | 100.1 | 55.0 | 18.6 | 96.4 |
> | **HalfCheetah-Medium-Expert** | 84.6 | 93.6 | 92.9 | 94.2 | 89.1 |
> | **HalfCheetah-Medium** | 44.6 | 47.9 | 43.9 | 48.2 | 45.9 |
> | **HalfCheetah-Medium-Replay** | 36.9 | 44.6 | 40.1 | 40.5 | 44.7 |
> | **Pen-Human** | 91.5 | 66.2 | 61.2 | 55.7 | 85.6 |
> | **Pen-clone** | 68.9 | 67.5 | 23.5 | 54.8 | 89.4 |
> | **Hammer-Human** | 1.1 | 18.4 | 1.1 | 1.2 | 3.7 |
> | **Hammer-clone** | 0.2 | 13.4 | 1.7 | 2.2 | 4.5 |
> | **Door-Human** | -0.1 | 0.1 | 0.2 | 0.7 | 17.1 |
> | **Door-Cloned** | 9.0 | 3.7 | 0.1 | 4.4 | 11.7 |
> | **Kitchen-Complete** | 60.0 | 22.5 | 16.3 | 27.5 | 79.3 |
> | **Kitchen-Mixed** | 47.5 | 30.0 | 45.0 | 60.0 | 59.5 |
> | **Kitchen-Partial** | 62.5 | 37.5 | 15.8 | 52.5 | 61.5 |
>
> The complete table is presented in Appendix I.1 (Page 24-25, Table 6) of the revised manuscript.
>
> **Q2. How coverage parameter σ is determined, and how well does the theoretical analysis based on this assumption correlate with actual performance?**
>
> We thank the reviewer for the insightful comment. Direct compute of coverage parameter $\sigma$  is theoretically and computationally infeasible for the following reasons: (a) By definition, $\Omega^\*\_{s\_{query}}$, consists of the transitions that would yield the optimal local linear approximation. However, these transitions may not exist in the offline dataset, and the dataset provides no oracle indicating which samples “should” belong to the ideal set. (b) The optimal local linear weight $w^\*\_s$ for solving $w\_s(\Omega^\*\_s)=w^\*\_s$ is unknown. (c) Even if $w^\*\_s$ is known and offline dataset $D$ contains all necessary ideal local transition sets, the combinatorial search for correct $\Omega^\*\_s$ is NP-Hard.
>
> Given these situations, our analysis focuses on how the empirical coverage induced by k-NN retrieval affects performance. Practically, the context length $k$ serves as a consistent proxy: a small $k$ indicates weak coverage, a moderate $k$ indicates reasonable coverage and a large $k$ may include extra noise. The coverage view offers useful intuition for how context length influences the effectiveness of local set, and our experiments in Section 4.3.2 match this intuition.

---

> ### Author Response · Authors · 2025-11-25
>
> **Q3. Failure case on Hammer-Human**
>
> We appreciate the reviewer highlighting this failure case. We found that Hammer-Human exhibits two properties that make it particularly challenging for ICQL:
>
> 1) Small dataset size and sparse coverage. Hammer-Human contains only 24 trajectories (\~11k transitions), vastly fewer than Hammer-Cloned (\~996k transitions). This leads to large distances between the query state and its retrieved neighbors that violate locality assumptions, and poor state-space coverage that make retrieval more likely to pull in semantically irrelevant transitions.
>
> 2) Low-quality transitions and noisy rewards. Most Hammer-Human trajectories have very low returns. So for each query state, the retrieved neighbors tend to have weak reward signals, making it more difficult to fit effective local Q-function.
>
> Although Hammer-Cloned contains mostly low-return behavior, the large dataset size provides much denser coverage. ICQL can retrieve states that are substantially closer to the query state, enabling more reliable local linear approximation and producing slightly higher scores.
>
> Moreover, we would like to note that for both Hammer-Human and Hammer-Cloned, the extremely low proportion of high-reward transitions makes it inherently difficult to retrieve any local neighborhood that provides strong positive supervision. As a result, even if the Q-network successfully fits a local linear approximation, it rarely observes transitions that reliably correspond to high-return behavior. Consequently, the learned Q-values cannot meaningfully distinguish truly rewarding actions, leading to uniformly low evaluation scores across both datasets.
>
> We provide dataset statistics comparisons below, and retrieval comparison is provided in the Appendix I.5 (Page 29) of the revised manuscript, both of which confirm these observations:
>
> | **Dataset** | **Hammer-Human** | **Hammer-Cloned** |
> | --- | --- | --- |
> | **Number of trajectories** | 24 | 3605 |
> | **Number of transitions** | 11285 | 996394 |
> | **Mean Trajectory Length (Minimum ~ Maximum)** | 455.2 (347-623) | 276.4 (199-623) |
> | **Mean Trajectory Return (Minimum ~ Maximum)** | 2817.5 (-109 ~ 16022) | 779.8 (-407 ~ 16022) |
>
> These findings also partially support our local-set coverage analysis: environments with insufficient local coverage constitute failure cases for ICQL. We have added this analysis to Appendix I.5 (Page 29) of the revised manuscript.
>
> **Q4. How does policy function without access to retrieval during evaluation? Does train–test mismatch occur?**
>
> We appreciate the opportunity to clarify. ICQL follows a standard actor–critic-like training paradigm where the critic uses retrieved context to estimate local Q-values and the policy learns from these Q-values. At evaluation time, only the policy is used, which is consistent with standard actor-critic practice.
>
> As our retrieval only directly enhances the critic part during training and critic is inherently not required during inference, this setup does not introduce train-test mismatch.
>
> ---
>
> We sincerely thank the reviewer again for the detailed feedback and encouraging evaluation. Your comments have helped us substantially improve the clarity, completeness, and positioning of our work. We hope our responses adequately address your concerns, and we would be delighted to provide further clarification or additional analysis if helpful.

---

> > ### Comment · Reviewer_MZEj · 2025-11-26
> >
> > Thank you for the response. I have several follow-up questions:
> >
> > 1. Does ICQL perform well in more complex sparse-reward environments, such as AntMaze? If so, could the authors elaborate on the underlying reasons for this effectiveness?
> >
> > 2. The supplementary experimental results do not appear to include standard deviations or confidence intervals. Could the authors provide these statistical measures to better assess the reliability and reproducibility of the reported results?
> >
> > 3. Have the authors investigated the impact of different distance metrics on the performance? An ablation study on this aspect would strengthen the empirical analysis.

---

> ### Author Response · Authors · 2025-12-04
>
> We thank the reviewer for the follow up questions. Below are our responses.
>
> **1. Does ICQL perform well in more complex sparse-reward environments, such as AntMaze? If so, could the authors elaborate on the underlying reasons for this effectiveness?**
>
> ICQL currently does not support AntMaze well for the following reason: the offline dataset of AntMaze contains a large proportion of failure or meaningless transitions, which make the retrieval quality too low for ICQL to learn meaningful local value function. Similar to the analysis on Hammer-Human and Hammer-Cloned, it rarely observes transitions that reliably correspond to high-return behavior by retrieval. Therefore, it is difficult for ICQL to help distinguish “good” and “bad” actions to learn in this case.
>
> **2. The supplementary experimental results do not appear to include standard deviations or confidence intervals. Could the authors provide these statistical measures to better assess the reliability and reproducibility of the reported results?**
>
> The standard deviations will be updated in the manuscript to confirm that the reported results are sufficiently reliable.
>
> **3. Have the authors investigated the impact of different distance metrics on the performance? An ablation study on this aspect would strengthen the empirical analysis.**
>
> We thank the reviewer for the kind suggestion. We have investigated in our Section 4.3.3 that different retrieval strategy does has impact on performance.

---

### Official Review · Reviewer_srbX · 2025-10-31

**Soundness:** 2
**Presentation:** 2
**Contribution:** 3
**Rating:** 6
**Confidence:** 3

**Summary:**

This paper proposes an offline RL algorithm called ICQL which solves offline Q-learning as a contextual inference problem. It retrieves local transitions from the dataset and fits a local linear Q-function via a linear-attention critic. The authors theoretically prove the bounded Q-approximation error and near-optimality guarantee for the greedy policy. In the experiments, ICQL outperforms most of the offline tasks in D4RL against offline RL baselines.

**Strengths:**

1. The authors provide a clear formulation of their method and theoretical justificaiton of Q-approximation error bound and performance gap bound. The theoretical contribution is solid. The idea is simple but quite interesting. Capturing the local structure is often easier than considering the global pattern. The authors gives a feasible approach to this direction.
1. The experiments on the well-known D4RL benchmark are comprehensive. They demonstrate a clear comparison against popular offline RL algorithms.
1. The supplementary material gives sufficient information for theory verificaiton and experiment verification. They are very helpful for follow-up research.

**Weaknesses:**

1. In offline RL, the coverage ratio $\sigma$ is an important factor influencing algorithm performance. In section 4.3.3, the authors only provide an indirect ablation to analyze the effect of $\sigma$. A direct analysis of the coverage of ideal local transition set would make the work more practical.
1. I notice some failure cases. While I often have a good tolerance on failure cases in novel methods, the performance on Hammer-Human-v1 looks too bad. A deepr analysis of this environment would be helpful to clarify the method's scope of applicability.
1. The Figure 3 does not make sense to me.
    1. Does SAC work as an oracle Q value estimator in this figure? If so, since SAC has two Q networks, which one do the authors use? Additionally, do all the three methods share the same policy network to ensure fairness? I also suggest approximating Q values via Monte Carlo rollouts using the same policy as the oracle, which is less biased by the training algorithms.
    1. The Q values should be normalized to the same scale across three plots so it can provide a uniformed color scheme to readers. The current plots are hard to analyze.
    1. I suggest replacing the tSNE plots with scatter plots that directly show the correlation of the estimated Q values and oracle Q values.
Minor issue
1. In line 377, it should be "Figure. 3" instead of a stand alone "3".

**Questions:**

1. The Definition 3.2 is also not clear to me
    1. In Equation 4, why is it $-||s_{query} - s_i||^2$? If I understand correctly, it should be $k$ nearest states to $s_{query}$, right?
    1. $d_{min}^s$ is the minimal distance to $s_{query}$ in $\overline{\Omega}^k_{s_{query}}$. Shouldn't $\Omega_{s_{query}}^{d_{min}}$ be empty since there is no closer states in this set?
1. The algorithm is still a bit unclear to me. Should the local weight vector $w_s^*$ be trained for every $s_{query}$? If so, I recommend the authors make a comprehensive analysis in training cost as it will take enormous computational resource.
1. Why does the algorithm fail on Hammer-Human-v1, could the authors make further justificaiton on this failure case?

---

> ### Author Response · Authors · 2025-11-25
>
> We sincerely thank the reviewer for the thoughtful and encouraging assessment of our paper. We greatly appreciate the recognition of our theoretical contributions, the clarity of our formulation, the comprehensiveness of our empirical evaluation, and the usefulness of our supplementary material. Below we address each question in detail.
>
> **W1. Analysis gap of coverage ratio $\sigma$**
>
> We thank the reviewer for the insightful suggestion. We agree that directly analyzing the coverage of the ideal local transition set $\Omega^\*\_{s\_{query}}$ would be valuable. Unfortunately, such direct analysis is theoretically and computationally infeasible for the following reasons: (a) By definition, $\Omega^\*\_{s\_{query}}$, consists of the transitions that would yield the optimal local linear approximation. However, these transitions may not exist in the offline dataset, and the dataset provides no oracle indicating which samples “should” belong to the ideal set. (b) The optimal local linear weight $w^\*\_s$ for solving $w\_s(\Omega^\*\_s)=w^\*\_s$ is unknown. (c) Even if $w^\*\_s$ is known and offline dataset $D$ contains all necessary ideal local transition sets, the combinatorial search for correct $\Omega^\*\_s$ is NP-Hard.
>
> Given these situations, our analysis focuses on how the empirical coverage induced by k-NN retrieval affects performance. The coverage view offers useful intuition for how context length influences the effectiveness of local set, and our experiments indeed show that different retrieval strategies lead to measurable performance differences.
>
> **W2&Q3. Analysis of failures in Hammer-Human**
>
> We appreciate the reviewer highlighting this failure case. We found that Hammer-Human exhibits two properties that make it particularly challenging for ICQL:
>
> 1) Small dataset size and sparse coverage. Hammer-Human contains only 24 trajectories (\~11k transitions), vastly fewer than Hammer-Cloned (\~996k transitions). This leads to large distances between the query state and its retrieved neighbors that violate locality assumptions, and poor state-space coverage that make retrieval more likely to pull in semantically irrelevant transitions.
>
> 2) Low-quality transitions and noisy rewards. Most Hammer-Human trajectories have very low returns. So for each query state, the retrieved neighbors tend to have weak reward signals, making it more difficult to fit effective local Q-function.
>
> Although Hammer-Cloned contains mostly low-return behavior, the large dataset size provides much denser coverage. ICQL can retrieve states that are substantially closer to the query state, enabling more reliable local linear approximation and producing slightly higher scores.
>
> Moreover, we would like to note that for both Hammer-Human and Hammer-Cloned, the extremely low proportion of high-reward transitions makes it inherently difficult to retrieve any local neighborhood that provides strong positive supervision. As a result, even if the Q-network successfully fits a local linear approximation, it rarely observes transitions that reliably correspond to high-return behavior. Consequently, the learned Q-values cannot meaningfully distinguish truly rewarding actions, leading to uniformly low evaluation scores across both datasets.
>
> We provide dataset statistics comparisons below, and retrieval comparison is provided in the Appendix I.5 (Page 29), both of which confirm these observations:
>
> | **Dataset** | **Hammer-Human** | **Hammer-Cloned** |
> | --- | --- | --- |
> | **Number of trajectories** | 24 | 3605 |
> | **Number of transitions** | 11285 | 996394 |
> | **Mean Trajectory Length (Minimum ~ Maximum)** | 455.2 (347-623) | 276.4 (199-623) |
> | **Mean Trajectory Return (Minimum ~ Maximum)** | 2817.5 (-109 ~ 16022) | 779.8 (-407 ~ 16022) |
>
> These findings also partially support our local-set coverage analysis: environments with insufficient local coverage constitute failure cases for ICQL.

---

> ### Author Response · Authors · 2025-11-25
>
> **W3. Clarification and improvements for Figure 3**
>
> 1. **Does SAC serve as the oracle Q, and which Q network is used?**
>
>     Yes. SAC is used as the oracle Q estimator. Following common practice, we use the minimum of the two SAC Q-networks to reduce overestimation bias, as in the training stage of SAC algorithm.
>
>     All three Q-networks are evaluated on the exact same rollout trajectory, ensuring that sampled (state, action) pairs are identical and not biased by training differences.
>
> 2. **Normalization of Q scales**
>
>     We thank the reviewer for this helpful suggestion. We agree that normalizing Q-values improves interpretability. We now scale all Q estimates into the [0,1] range before visualization. We have updated the figure in our manuscript accordingly. Now Figure 3 clearly shows the higher similarity between SAC estimates and ICQL estimates, supporting our observation that ICQL can produce a more similar value estimation to effective online RL methods than IQL.
>
> 3. **Suggestion to use scatter plots**
>
>     We appreciate this excellent suggestion. We now include additional scatter plots comparing each method’s estimated Q-values against the SAC oracle.
>     These plots make correlation patterns clearer and complement the t-SNE visualizations.
>
>  We have updated these plots to Appendix I.7 (Page 31-32).
>
> **W4. Minor correction on Figure 3 reference.**
>
> Thank you for catching this. We have corrected the reference in the revised manuscript.
>
> **Q1. Clarifying Definition 3.2**
>
> 1. **In Equation 4, why is it $-||s\_{query}-s\_i||^2$? If I understand correctly, it should be $k$ nearest states to $s\_{query}$, right?**
>
>     Yes, that interpretation is correct. The set refers to the k nearest states to the query state.
>
> 2. **$d^s\_{min}$ is the minimal distance to $s\_{query}$ in $\bar{\Omega}^k\_{s\_{query}}$. Shouldn't** $\Omega^{d\_{min}}\_{s\_{query}}$ **be empty since there is no closer states in this set?**
>
>     We thank the reviewer for pointing out the ambiguity. Here, $\Omega^{d\_{min}}\_{s\_{query}}$ is not intended to denote the set of states strictly closer than $d\_{min}$. Instead, it denotes the retrieved top-k neighborhood whose minimal distance is $d\_{min}$. To avoid this confusion, we will replace $d\_{min}$ with $d\_k$ and rename the set as $\Omega^{d\_k}\_{s\_{query}}$ to more clearly indicate that it corresponds to the k-nearest-neighbor radius.
>
> **Q2. Clarify whether the local weight vector $w^\*_s$ be trained for every $s\_{query}$ and recommend a comprehensive analysis in training cost.**
>
> Yes, the local weight vector $w^\*\_s$ is computed for every $s\_{query}$. This learning process only needs a forward pass through the linear transformer, in which the $w\_s$ will be optimized with in-context learning. During training, the back-propagation does not directly involve $w\_s$.
>
> We provide compute measurements for varying context lengths (K = 10, 20, 30, 40) for the Walker2d tasks.
>
> | **Dataset** | **Train K=10(ms)** | **Train K=20(ms)** | **Train K=30(ms)** | **Train K=40(ms)** |
> | --- | --- | --- | --- | --- |
> | **Medium-Expert** | 48.90 | 70.73 | 111.75 | 170.71 |
> | **Medium** | 46.63 | 71.77 | 113.82 | 170.85 |
> | **Medium-Replay** | 48.68 | 74.75 | 115.43 | 171.97 |
>
> Additionally, GFLOPs and peak memory comparisons across different context lengths and number of linear attention layers are provided to give a more complete view of computational overhead, where the number of linear attention layers corresponds to the number of updates for learning each $w_s$.
>
> GFLOPs Comparison:
> | # Layers | K=10 | K=20 | K=30 | K=40 |
> | --- | --- | --- | --- | --- |
> | **10** | 0.25 | 0.51 | 0.81 | 1.14 |
> | **20** | 0.50 | 1.03 | 1.62 | 2.28 |
> | **30** | 0.75 | 1.54 | 2.43 | 3.42 |
> | **40** | **1.00** | **2.06** | **3.24** | **4.56** |
>
> Peak Memory Consumption:
> | # Layers | K=10 | K=20 | K=30 | K=40 |
> | --- | --- | --- | --- | --- |
> | **10** | 171.28 | 306.56 | 445.57 | 590.39 |
> | **20** | 209.71 | 375.38 | 549.51 | 738.00 |
> | **30** | 248.39 | 443.29 | 655.26 | 879.30 |
> | **40** | 288.94 | 511.58 | 758.94 | **1023.75** |
>
> We have added these evaluations to Appendix I.4 (Page 26-28) in the revised manuscript.
>
> ---
>
> We thank the reviewer again for the constructive and encouraging feedback. Your comments have helped improve both the clarity and rigor of our theoretical and empirical presentation. We hope that the above clarifications address your concerns, and we would be very happy to discuss any additional questions or suggestions.

---

### Official Review · Reviewer_qZiC · 2025-10-31

**Soundness:** 3
**Presentation:** 3
**Contribution:** 3
**Rating:** 6
**Confidence:** 3

**Summary:**

The paper introduces In-Context Compositional Q-Learning (ICQL) for offline RL: a transformer retrieves top-k similar transitions and fits a local linear Q for each query, instead of learning a single global critic. Claims: (i) with bounded features and adequate retrieval coverage, the greedy policy is near-optimal (performance-gap bound); (ii) on D4RL (Mujoco/Adroit/Kitchen), ICQL yields consistent gains—largest on compositional tasks—supported by ablations on context length, transformer depth, and retrieval strategy. The work positions ICQL as retrieval-conditioned value learning (not return-conditioned action modeling), aiming to better exploit task locality than standard offline RL critics.

**Strengths:**

Clear idea & scope: reframes Q-learning as local, retrieval-conditioned regression—intuitive and timely.
Theory with knobs: bound ties performance to local linearity and retrieval coverage, matching design levers.
Empirically persuasive where it should be: biggest wins on multi-stage “Kitchen,” aligned with the compositionality thesis.
Ablations with signal: context length / retrieval choices matter in the expected directions.

**Weaknesses:**

Baseline gap: no direct comparison to retrieval-augmented sequence models (e.g., Retrieval-Augmented Decision Transformer: External Memory for In-context RL). Both ICQL and retrieval-augmented Decision Transformer methods use external memory to condition decisions in offline RL

Compute transparency: training-time cost/latency vs. IQL/CQL/TD3+BC not reported.

Retrieval proxy: L2 proximity in state space may misalign with Q-similarity; robustness to metric choice is under-explored.

**Questions:**

"We observe that, for each RL control task, the state space can be inherently divided into multiple sub-tasks": what kind of sub-tasks, please add some background description? Different parts of the state space correspond to different “phases”

Figure1: I don't see obvious difference between QueryB-R1,2,3.

Have you tried learned value-aware similarity (contrastive encoders predicting Q-proximity) or state–action NN retrieval, and how does that affect the theory’s coverage term?

Can you report FLOPs/memory vs. baselines and show compute–performance trade-offs for context length?

---

> ### Author Response · Authors · 2025-11-25
>
> We sincerely thank the reviewer for the thoughtful and constructive feedback. We appreciate the clear articulation of strengths and the helpful suggestions for additional analyses and comparisons. Below we address each point in detail.
>
> **W1. Baseline gap: comparison to retrieval-augmented sequence models**
>
> Thank you for this valuable suggestion. Following the reviewer’s advice, we conducted additional experiments comparing ICQL and RADT across the D4RL benchmark. The results are shown below, where ICQL consistently outperforms RADT on most of the tasks. We have added these results to Appendix I.1 (Page 24-25) of the revised manuscript.
>
> | **Task** | **RADT** | **ICQL** |
> | --- | --- | --- |
> | **Walker2d-Medium-Expert** | 107.8 | **113.3** |
> | **Walker2d-Medium** | 68.9 | **80.3** |
> | **Walker2d-Medium-Replay** | 67.2 | **81.9** |
> | **Hopper-Medium-Expert** | 109.4 | **113.3** |
> | **Hopper-Medium** | 62.4 | **62.6** |
> | **Hopper-Medium-Replay** | 81.6 | **96.4** |
> | **HalfCheetah-Medium-Expert** | **90.9** | 89.1 |
> | **HalfCheetah-Medium** | 42.0 | **45.9** |
> | **HalfCheetah-Medium-Replay** | 38.9 | **44.7** |
> | **Pen-Human** | 17.8 | **85.6** |
> | **Pen-clone** | 32.4 | **89.4** |
> | **Hammer-Human** | 0.7 | **3.7** |
> | **Hammer-clone** | 1.3 | **4.5** |
> | **Door-Human** | 13.2 | **17.1** |
> | **Door-Cloned** | 2.4 | **11.7** |
> | **Kitchen-Complete** | 32.5 | **79.3** |
> | **Kitchen-Mixed** | 54.1 | **59.5** |
> | **Kitchen-Partial** | 53.8 | **61.5** |
>
> **W2&Q4. Compute transparency: training-time cost, FLOPs, memory, and context-length trade-offs**
>
> We appreciate this insightful suggestion. We have added a comprehensive compute analysis across all baselines on Walke2d-Medium-Expert dataset, as summarized below:
>
> | Algorithm | Avg Time per Training Step (ms) | Avg Time per Inference Step (ms) | Training FLOPs (GFLOPs) | Peak Memory (MB) |
> | --- | --- | --- | --- | --- |
> | TD3BC | 7.23 | 0.26 | 0.17 | 30 |
> | IQL | 10.52 | 0.61 | 0.22 | 26 |
> | CQL | 47.57  | 0.61 | 2.64 | 79 |
> | DT | 68.42 | 2.89 | 151.40 | 1383 |
> | RA-DT | 121.02 | 3.13 | 1103.79 | 1424 |
> | ReBRAC | 13.91 | 0.26 | 0.18 | 38 |
> | DMG | 32.33 | 0.42 | 0.55 | 27 |
> | FQL | 19.63 | 0.37 | 4.53 | 126 |
> | QC | 21.60 | 0.25 | 4.65 | 244 |
> | ICQL | 70.73 | 0.51 | 1.03 | 375 |
>
> We further report training time consumption for varying context lengths in $\{10,20,30,40\}$:
>
> | **Context length** | **Avg Time per Training Step (ms)** |
> | --- | --- |
> | **10** | 48.90 |
> | **20** | 70.73 |
> | **30** | 111.75 |
> | **40** | 170.71 |
>
> This analysis shows that while ICQL incurs moderate additional compute cost relative to most advanced baselines, and it remains more efficient than sequential models (DT/RADT) while achieving substantially stronger performance. The training time needed scales with context length, and using a context length of 20 remains comparable efficient while providing competitive performance. All the results have been added to the Appendix I.4 (Page 26-28).
>
> **W3. Alignment of L2 proximity with Q-similarity and metric robustness**
>
> We thank the reviewer for raising this important point. We completely agree that the alignment between state-space proximity and Q-similarity is not perfect, and the choice of the retrieval metric is a crucial aspect of retrieval-based RL methods. However, we note that the objective of retrieval in our method is not to find states with similar Q-values, but to construct a local dataset of relevant transitions that are dynamically and semantically related to the query state. As the reviewer implied, this includes both successful and unsuccessful experiences. Even when Q-values differ, nearby states often share similar short-horizon dynamics, enabling more stable temporal-difference regression and reducing extrapolation error.
>
> The core aim of this paper is to establish a simple yet effective in-context Q learning algorithm with retrieval mechanism. The investigation of optimal retrieval metrics is orthogonal to the core contribution and represents a promising direction for future work.

---

> ### Author Response · Authors · 2025-11-25
>
> **Q1. Clarifying “sub-tasks” in the state space**
>
> Thank you for raising this point. To clarify the motivation, we first refer to Figure 1 in the paper. In this visualization, Query A and its nearest neighbors correspond to a sub-task to **recover from a backward-leaning pose**. These states represent the agent losing balance and performing corrective motions to stabilize itself. In contrast, Query B and its nearest neighbors correspond to a sub-task to **keep stable high-speed forward locomotion**. These states lie on a smooth part of the trajectory where the agent maintains momentum and forward velocity.
>
> These examples illustrate that within a single locomotion task, the state space naturally decomposes into semantically coherent sub-regions that correspond to different behavioral phases (e.g., recovery, propulsion, stabilization). States within the same local region share consistent short-horizon dynamics, making them suitable for constructing a local dataset for value estimation.
>
> Another example is the **Kitchen** environment, which is inherently a multi-task, multi-stage setting: the agent must complete several independent manipulation subtasks (e.g., opening the microwave, flipping the switch, moving the kettle, etc.). Each subtask occupies a distinct portion of the state space, reflecting different underlying dynamics and reward structures. This further supports our motivation that leveraging nearby states provides behaviorally coherent context for estimating the local Q-function.
>
> **Q2. Figure1: difference between QueryB-R1,2,3.**
>
> We would like to note that this similarity is expected, and it and reflects the goal of the retrieval mechanism: to obtain through state-based retrieval. To make this explicit, we now report their full (state, action, reward) triplets, reproduced below:
>
> Query B:
>
> - State: [1.176, 0.234, -0.013, 0.034, 0.907, 0.038, 0.015, 0.102, 4.699, -0.115,
> 3.589, 8.436, -6.821, -7.925, 0.057, -0.242, -5.536]
> - Action: [0.115, 0.971, 0.915, 0.990, -0.993, -0.906]
> - Reward: 5.702
>
> R1:
>
> - State: [1.175, 0.243, -0.002, 0.027, 0.886, 0.042, 0.012, 0.104, 4.571, -0.093,
> 3.161, 8.085, -5.472, -7.117, 0.002, -1.169, -2.745]
> - Action: [-0.089, 0.993, 0.960, 0.739, -0.982, -0.810]
> - Reward: 5.572
>
> R2:
>
> - State: [1.168, 0.282, 0.007, 0.038, 0.905, 0.042, 0.009, 0.029, 4.461, -0.140,
> 3.190, 8.152, -5.003, -6.946, 0.078, -1.357, 0.181]
> - Action: [-0.493, 0.993, 0.888, 0.974, -0.996, -0.923]
> - Reward: 5.470
>
> R3:
>
> - State: [1.173, 0.243, -0.017, 0.036, 0.899, 0.038, 0.003, 0.016, 4.455, -0.170,
> 2.809, 7.572, -5.325, -7.364, 0.030, -2.101, 0.873]
> - Action: [-0.358, 0.997, 0.939, 0.326, -0.994, -0.807]
> - Reward: 5.463
>
> All three retrieved transitions closely resemble the query, confirming that the retrieval mechanism is functioning as intended.
>
> **Q3. Discussion on learned value-aware similarity or state–action NN retrieval, and how does that affect the theory’s coverage term?**
>
> We agree with the reviewer that this is a great idea to retrieve by similar Q-value to augment query value and action prediction. This method will theoretically increase coverage ratio and improve the relevance of retrieved transitions. However, as the methodology of our method is to utilize state-similar transitions for local Q-value estimation, applying such learned metrics would introduce substantial modeling complexity and tuning requirements. Given our goal of establishing the simplest instantiation of in-context retrieval-based Q-learning, we leave this interesting extension to future work.
>
> ---
>
> We sincerely appreciate the insightful feedback, which has meaningfully strengthened both the empirical and theoretical presentation of our work. We hope that our responses help clarify each of the reviewer’s concerns, and we would be happy to engage in further discussion or address any remaining questions.

---

### Official Review · Reviewer_YKZA · 2025-11-01

**Soundness:** 3
**Presentation:** 2
**Contribution:** 2
**Rating:** 4
**Confidence:** 4

**Summary:**

This paper proposes ICQL, an offline RL method that reframes Q-learning as contextual inference. For each query state-action it retrieves a small, state-similar set of transitions from the offline dataset and uses a linear transformer to infer a local linear Q-function, then plugs this into an IQL-style training pipeline. The motivation is that offline datasets often have locally coherent regions of value, but adjacent regions can differ sharply. A single global critic tends to smooth these away, whereas ICQL tries to learn "per-query" value estimates from retrieved context. On D4RL benchmarks, ICQL often outperforms strong offline RL baselines and the ablations support the claim that retrieval/local context actually matters.

**Strengths:**

- Treats offline Q-learning as "retrieve local neighbors, do in-context TD, act", which matches the empirical observation that value structure is local and nonuniform across the dataset.
- ICQL often improves over IQL/CQL-style baselines on MuJoCo.
- Varying context length / $k$ and retrieval strategy shows performance is sensitive to the retrieved neighborhood, which supports the paper's core claim that which transitions you condition on matters.
- The paper states assumptions (local linear approximability, coverage of the local set) and derives a performance bound.

**Weaknesses:**

- ICQL requires a retrieval step for every training query. For large offline buffers this $k$-NN step can dominate wall-clock, but the paper does not report complexity, index structure (exact vs. ANN), or training time vs. IQL/CQL. This is a real deployment concern the paper currently ignores.

- The method’s notion of a "nearby transition" is Euclidean distance in the (low-dim) state space. This is reasonable for MuJoCo/Adroit/Kitchen but does not address high-dimensional observations (images) where $L_2$ is brittle and the coverage assumption becomes much harder to satisfy. A learned or task-aware retriever is not explored, only mentioned.

- The theory leans on "within a small neighborhood the Q-function is linearly approximable". This is a strong assumption for tasks with multimodal returns, contact-rich transitions, or poorly aligned state features. The paper does not analyze what happens when the neighborhood is not well modeled by a single linear head.

- The current wording about "nearby clusters presenting as noise" is hard to follow and mixes an empirical visualization (clusters in the dataset) with the formal local-set definition.

- There are experiments varying context length / $k$, and they do show non-monotonic behavior, but the paper avoids discussing the $d,\bar d$ definitions it introduces earlier and effectively hides the choice inside $k$-NN. A clearer statement of how performance scales with $k$ (and how they pick it) would help.

- The paper instantiates the method with a linear transformer/head but does not compare against a slightly more expressive local model (e.g., a small MLP on the retrieved set). As a result, it is unclear whether "in-context linear-TD" is essential, or just convenient.

- There are typos and phrasing that make already technical sections (the definitions and the "in-context TD" explanation) harder to read.

**Questions:**

- What happens in ICQL when the retrieved neighborhood is not well modeled by a single linear head (e.g. contact-rich or multimodal regions)?

- Could the authors clarify what is meant by "nearby clusters presenting as noise" and how this maps to the formal local-set definition?

- The authors introduce $(d,\bar d)$ for local sets, but the experiments effectively tune $k$ in $k$-NN. How should we relate the theory parameters to the practical $k$?

- Is the choice of a linear transformer/head essential for stability/theory, or would a slightly more expressive local model (small MLP over the retrieved set) also work?

---

> ### Author Response · Authors · 2025-11-25
>
> We sincerely thank the reviewer for the detailed and constructive feedback. We appreciate the positive assessment of our empirical findings and theoretical contributions. Below, we address each question and clarify the motivations behind our design choices.
>
> **W1. Complexity concerns of per-query retrieval and training time comparisons.**
>
> Thank you for highlighting this important point. We fully agree that retrieval cost is a practical concern for deployment. To mitigate repeated computation, we **pre-compute all retrieval indices once before training**, since:
>
> 1. The offline dataset is fixed.
> 2. The retrieval rule is deterministic.
> 3. Pre-computation does not affect the learning dynamics or outcomes.
>
> This turns per-step retrieval cost into an amortized constant-time lookup during training.
>
> We use an **exact index** for reproducibility. The table below reports the real-time retrieval time, the lookup time with cached indices, and the training speed. The results are averaged across all datasets used in our experiments. The full result table is shown in the appendix of the manuscript. As shown, **cached retrieval adds only ~0.03 ms per step**, which is negligible relative to the overall training time.
>
> | |**Time (ms)**|
> |-|-|
> |**Retrieval with Cached Index**|0.03|
> |**Train with K=10**|46.94|
> |**Train with K=20**|72.15|
> |**Train with K=30**|113.86|
> |**Train with K=40**|171.95|
> |**Inference**|0.54|
>
> We also compare the average time for each training and inference step across baselines.
>
> |**Algorithm**|**Avg Time per Training Step(ms)**|**Avg Time per Inference Step(ms)**|
> |-|-|-|
> |**TD3BC**|7.23|0.26|
> |**IQL**|10.52|0.61|
> |**CQL**|47.57|0.61|
> |**DecisionTransformer**|68.42|2.89|
> |**RA-DT**|121.02|3.13|
> |**ReBRAC**|13.91|0.26|
> |**DMG**|32.33|0.42|
> |**FQL**|19.63|0.37|
> |**QC**|21.60|0.25|
> |**ICQL**|70.73|0.51|
>
> This analysis shows that while ICQL incurs moderate additional compute cost relative to most advanced baselines, and it remains more efficient than sequential models (DT/RADT) while achieving substantially stronger performance. The training time needed scales with context length, and using a context length of 20 remains comparable efficient while providing competitive performance. We have added these evaluations to Appendix I.4.3 (Page 27-28) in the revised manuscript.
>
> **W2. Applicability to high-dimensional observations & retrieval metrics**
>
> We appreciate this insightful observation. We agree that for image-based observations, Euclidean distance on raw states is inefficient, and learning a task-aware representation would likely improve retrieval quality.
>
> We would like to note that our intention in this paper is to isolate and evaluate the compositional in-context Q-learning mechanism, without additional confounding components such as learnable retriever. Therefore, we kept the experimental setup minimal and used the raw state vector to ensure a clean comparison against standard algorithms.
>
> To clarify in the manuscript, we now explicitly state that ICQL is agnostic to the distance metric, a learned retriever is a natural extension, and applying ICQL to image-based RL is a promising direction for future work.
>
> **W3&Q1. What happens in ICQL when the retrieved neighborhood is not well modeled by a single linear head (e.g. contact-rich or multimodal regions)?**
>
> We appreciate the opportunity to clarify this point. Our assumption does **not** require linearity in the raw state-action input space. Instead, ICQL assumes that the Q-function is locally linear in a latent representation learned by the encoder. This aligns with the structure of modern deep Q-networks: the last layer is a linear projection of the preceding nonlinear representation. Therefore, ICQL remains feasible even in multimodal or contact-rich regimes.
>
> **W4&Q2. Clarification of "nearby clusters presenting as noise" and how this maps to the formal local-set definition**
>
> Thank you for pointing this out. Our intention was to describe the following phenomenon observed in the data: although states in the dataset can be grouped into coherent clusters, where each typically corresponding to a specific subtask, two clusters that appear geometrically may nevertheless correspond to semantically different behaviors and exhibit distinct long-horizon returns.
>
> Therefore, for a query state belonging to cluster A, the transitions within cluster A form a meaningful local set for estimating its local Q-function. In contrast, transitions from a neighboring but semantically different cluster B do not share the same local value structure; including them in the local regression acts as noise rather than useful signal.
>
> This motivates the use of retrieval-based local neighborhoods: the goal is not to approximate a globally smooth Q-function, but to identify a subset of transitions that reflect the same underlying subtask dynamics for the query state.
>
> To avoid this ambiguity, we have rephrased the corresponding sentence in the manuscript.

---

> ### Author Response · Authors · 2025-11-25
>
> **W5&Q3. Relation between theory parameters $(d, \bar d)$ and practical hyper-parameter $k$**
>
> We thank the reviewer for this insightful comments. We clarify the connection between the theoretical formulation of $(d,\bar d)$ and $k$ as follows:
>
> In the theory, a local set $\Omega_{s_{query}}^d$ is defined as all transitions whose states fall within a radius-$d$ neighborhood around $s$. This radius determines the intrinsic “locality scale’’ at which the Q-function is assumed to be approximately linear. However, in practice, the radius $d$ is not directly tunable: it depends on the underlying density and geometry of the dataset and is unknown to the algorithm.
>
> Instead, ICQL controls locality through the retrieval size $k$. Retrieving the top-$k$ nearest neighbors is equivalent to selecting a data-adaptive radius, where $d_k=\max_{(s_i,\cdot)\in\text{top-}k}||s_i-s||^2_2$ and $\bar{d}\_k=\max\_{(s\_i,\cdot)\in\text{top-}k}||s'\_i-s'||^2\_2$, so that the practical neighborhood is exactly the theoretical local set with radius $(d_k,\bar{d}_k)$. A visualization analysis is provided in Appendix I.6 (Page 30).
>
> Thus, $k$ determines the effective radius implicitly and monotonically: larger $k$ expands the radius $(d_k,\bar{d}_k)$ and increases the size and heterogeneity of $\Omega^{d_k}_s$, while smaller $k$ leads to tighter neighborhoods with more consistent local value structure.
>
> We have added a clarification in the revised manuscript.
>
> **W6&Q4. Comparison against other local model on the retrieved set**
>
> We appreciate the reviewer for encouraging this ablation. We performed additional experiments replacing the linear transformer with other architectures, which is either a small MLP or a standard transformer. The following table show the results. The results demonstrate that the linear in-context mechanism is not only theoretically convenient for but also empirically essential for learning local Q function. We have added these results to the revised Appendix I.3 (Page 26).
>
> | **Task** | **Linear Attention** | **Linear MLP** | **Standard Attention** |
> | --- | --- | --- | --- |
> | **Walker2d-Medium-Expert** | **113.3** | 109.5 | 108.8 |
> | **Walker2d-Medium** | **80.3** | 76.7 | 77.4 |
> | **Walker2d-Medium-Replay** | **81.9** | 60.2 | 42.9 |
> | **Hopper-Medium-Expert** | **113.3** | 109.9 | 70.3 |
> | **Hopper-Medium** | **62.6** | 55.7 | 61.9 |
> | **Hopper-Medium-Replay** | **96.4** | 89.9 | 42.1 |
> | **HalfCheetah-Medium-Expert** | **89.1** | 83.0 | 72.5 |
> | **HalfCheetah-Medium** | **45.9** | 43.3 | 42.0 |
> | **HalfCheetah-Medium-Replay** | **44.7** | 39.2 | 36.1 |
> | **Pen-Human** | **85.6** | 66.6 | 72.7 |
> | **Pen-clone** | **89.4** | 80.7 | 83.8 |
> | **Hammer-Human** | 3.7 | **6.1** | 4.2 |
> | **Hammer-clone** | 4.5 | **7.9** | 1.8 |
> | **Door-Human** | **17.1** | 6.9 | 8.9 |
> | **Door-Cloned** | **11.7** | 3.5 | 3.4 |
> | **Kitchen-Complete** | **79.3** | 70.0 | 78.3 |
> | **Kitchen-Mixed** | **59.5** | 57.5 | 55.8 |
> | **Kitchen-Partial** | **61.5** | 48.3 | 55.8 |
>
> **W7. Typos and phrasing**
>
> We have carefully corrected all identified typos and improved phrasing throughout the manuscript.
>
> ---
> We sincerely thank the reviewer again for the thoughtful comments and constructive suggestions. Your feedback has helped us significantly clarify both the theoretical framing and the practical aspects of ICQL. We hope these clarifications resolve your concerns. If there any remaining questions, we would be happy to discuss.

---

### Meta-Review · Area_Chair_mEpP · 2025-12-12

**Summary:**

This paper proposes a novel retrieval based approach to enhance Q value estimation in offline RL. The key idea is to retrieve data similar to the current (s, a) pair from the offline dataset via kNN. Then in-context learning is performed through linear transformer using the retrieved data as context to gain a better local approximation of the Q value. Theoretical analysis is provided extending previous works about implementing linear TD with linear transformer. Extensive empirical results demonstrate the desired performance improvements.

Reviewers raise quite some concerns, mostly about the wall-clock time and the use of linear attention. I think the authors did a good job in providing new experiments to address the concerns about wall-clock time. The use of linear attention is a standard practice by theorists so it is not a problem to me. The authors even demonstrate that linear attention even outperforms standard attention, which in my opinion is a very interesting result. I went through all the reviews and responses. I did not spot any major flaw in reviews and most concerns are cleared  by response. MZEj may still have some uncleared concerns but at this point I feel it's pretty clear that the strengths of the work overweights the possible weakness. I, therefore, recommend accept.

**Reviewer Concerns:**

see summary above

**Reviewer Scores:**

YKZA may increase to 6. qZiC may increase to 8. srbX may remain 6. MZEj may remain 4.

---

### Decision · Program_Chairs · 2026-01-26

Accept (Poster)